# *In silico* analysis and *in planta* production of recombinant *ccl21/IL1β* protein and characterization of its *in vitro* anti-tumor and immunogenic activity

**Hasan Marashi**[1], **Maria Beihaghi**[1,2,3]*, **Masoud Chaboksavar**[2], **Samad Khaksar**[3], **Homan Tehrani**[4], **Ardavan Abiri**[5]

**1** College of Agriculture, Ferdowsi University of Mashhad, Mashhad, Iran, **2** Department of Biology, Kavian Institute of Higher Education, Mashhad, Iran, **3** School of Science and Technology, The University of Georgia, Tbilisi, Georgia, **4** Department of Paediatric, School of Medicine, Sabzevar University of Medical Sciences, Sabzevar, Iran, **5** Department of Medicinal Chemistry, School of Pharmacy, Kerman University of Medical Sciences, Kerman, Iran

* maria_beihaghi@yahoo.com

**Data Availability Statement:** All relevant data are within the manuscript.

## Abstract

CCL21 has an essential role in anti-tumor immune activity. Epitopes of IL1β have adjuvant activity without causing inflammatory responses. CCR7 and its ligands play a vital role in the immune balance; specifically, in transport of T lymphocytes and antigen-presenting cells such as dendritic cells to the lymph nodes. This study aimed to produce epitopes of CCL21 and IL1β as a recombinant protein and characterize its *in vitro* anti-tumor and immunogenic activity. A codon-optimized *ccl21/IL1β* gene was designed and synthesized from human genes. Stability and binding affinity of CCL21/IL1β protein and CCR7 receptor were examined through *in silico* analyses. The construct was introduced into *N. tabacum* to produce this recombinant protein and the structure and function of CCL21/IL1β were examined. Purified protein from transgenic leaves generated a strong signal in SDS PAGE and western blotting assays. FTIR measurement and MALDI-TOF/TOF mass spectrography showed that *ccl21/IL-1β* was correctly expressed in tobacco plants. Potential activity of purified CCL21/IL1β in stimulating the proliferation and migration of MCF7 cancer cell line was investigated using the wound healing method. The results demonstrated a decrease in survival rate and metastasization of cancer cells in the presence of CCL21/IL1β, and $IC_{50}$ of CCL21 on MCF7 cells was less than that of non-recombinant protein. Agarose assay on PBMCsCCR7+ showed that CCL21/IL1β has biological activity and there is a distinguishable difference between chemokinetic (CCL21) and chemotactic (FBS) movements. Overall, the results suggest that CCL21/IL1β could be considered an effective adjuvant in future *in vivo* and clinical tests.

**Funding:** The author(s) received no specific funding for this work.

**Competing interests:** The authors have declared that no competing interests exist.

## Introduction

Cytokines are small proteins crucial to the cell signaling of the immune system, affecting the function and behavior of the cells around them. They include lymphokines, interferons, tumor necrosis factors, chemokines, and interleukins [1]. Chemokines activate inflammatory pathways and direct leukocyte migration [2, 3]. CC chemokine ligand 21 (CCL21) is a chemokine without inflammatory responses, which attracts normal immune cells and metastasizing tumor cells to lymph nodes by activating the CCR7 receptor [4]. CCR7 is expressed on many cancer tumor cells and naïve T-cells [5]. Moreover, CCL21 is a recognized attractant for CCR7-positive (CCR7+) cells [6]. Therefore, CCL21/CCR7 has essential roles in lymphatic cancer metastasis, immune cell and lymph-node homing, peripheral tolerance, development, and function of T regulatory cells in such diseases as HIV and COVID-19 [5], lymphoid neogenesis [7, 8], and in secondary lymphoid organs. Activation of T-cells is facilitated by CCL21 through recruitment and co-localization of naïve lymphocytes and antigen-induced dendritic cells (DC) [3, 9, 10]. Accordingly, CCL21 is a base for cancer immunotherapy since it can chemoattract T-lymphocytes and DCs [11]. DCs take up tumor antigens and migrate to T-cell zones of lymphoid organs for specific anti-tumor T-cell activity [7, 12]. Examples of CCR7 + human breast cancer cell lines are MCF7 and MDA-MB232 [13]. In addition to its anti-tumor activity, CCL21 is also effective in treating AIDS. HIV attaches to chemokine receptors such as CCR7 on the surface of CD4+ T-cells and kills immune cells. However, studies have shown that CCL21 is a chemokine that can occupy CCR7 receptors on the surface of CD4+ T-cells and prevent HIV attachment [14].

Interleukin-1 beta (IL-1β) is a potent inflammatory cytokine protein encoded by the *IL1β* gene in humans. IL-1β is also involved in various cellular activities such as activation of neutrophils, B-cells and T-cells, production of cytokines and antibodies, proliferation of fibroblasts, and collagen production [15]. The VQGEESNDK amino acid sequence is a part of IL-Iβ that has all the adjuvant activity of the entire IL-Iβ, without the inflammation and fever response. Consequently, by inserting the VQGEESNDK sequence into the synthetic construct, the immunogenicity of protein antigens is increased [16].

Major histocompatibility complex (MHC) is a part of the genome that plays an essential role in creating an immune response to protein antigens. MHC genes produce the MHC molecules. These molecules can bind to some antigenic peptides, after which the specific complex of MHC peptide will be detectable to T-lymphocytes [17]. For example, CCL21 is a Chemokine that controls cell trafficking and is involved in numerous pathologic and inflammatory conditions, and it is endocytosed with its receptor (CCR7) via both MHC class I and II processing pathways to induce CD8+ and CD4+ T-cell responses [18].

The use of plants as bioreactors for manufacturing recombinant proteins, known as molecular farming, has captured great interest among researchers over the past two decades [19]. Tobacco (*Nicotiana tabacum* L.) serves as one of the model plants used in physiological, genetic, and tissue-culture studies [20, 21]. Manipulation of the tobacco genome requires efficient transformation and regeneration techniques, making this plant an impressive bioreactor for production of recombinant proteins. Tobacco tissue culture system [20] and transformation method were optimized in the present study. Several strategies have been proposed for boosting recombinant protein expression, including the use of influential promoters, chloroplast transformation [22, 23], signal peptide codon optimization, and application of untranslated leader sequences [24]. Molecular dynamics simulation [25] is a computer simulation method used to understand the conformational changes caused by mutations in recombinant proteins [26, 27]. In addition, it can be used to analyze the highly fluctuating and complex dynamics of proteins [28].

This study aimed to investigate the expression of *ccl21/IL1β* as recombinant protein in transgenic lines. So, in addition to the CCL21 sequence, this recombinant protein contains fragments of interleukin 1 beta sequences, introduced by placing epitopes related to B-lymphocytes and CTLs in the desired structure to activate T & B-lymphocytes, macrophages, and neutrophils simultaneously. It can be utilized as a potent adjuvant for treating breast cancer, especially with Her2 / neu gene expression and the Th1 immune response, which affects most drugs, alleviates the disease, and boosts immunity. Therefore, the gene construct to be expressed in tobacco plants contained a DNA fragment encoding 134 amino acids that form epitopes and which cover more than 90% of the HLA-DRB4 allele and can bind to MHCI and MHCII molecules at T-cell and CCR7 receptors. *In silico* analysis and *in vitro* assays (such as scratch assay, MTT assay, and chemotaxis assay) were used for investigating the immunogenic function of this recombinant protein. The project was ethically approved in accordance with Code IR.MUMS.REC.1399.428.

## 2. Materials and methods

### 2.1 *In silico* analysis

**2.1.1 Construction of expression cassette.** Protein sequences encoding human CCL21 (Accession No: O00585) had a DCCL motif, a domain binding to the CCR7 receptor and VQGEESNDK sequence; a part of selected epitopes of IL-1β is involved in inflammatory and immune responses and has high adjuvant effects, were designed as the principal part of the expression cassette. The designed expression cassette included Kozak consensus sequence as the plant-specific ribosome binding site (GCCACC). The CCL21 and VQGEESNDK amino sequences of human IL-1 beta were linked together by EAAAK sequence as linker that reduces interaction with other recombinant protein domains. The SEKDEL signal sequence—for effectual agglomeration of the recombinant protein in the endoplasmic reticulum (ER)—and 6xHis tags for purification of the gene construct were also linked by RVLAEA sequence as HIV protease linker respectively. Recognition sites of *Bam*HI and *Sac*I restriction enzymes were added to the 5' and 3' ends of the synthetic gene, respectively. AUG and UAA were also added to the construct's 5' and 3' ends as start and stop codons, respectively. The terminal part of the chimeric gene was optimized based on the codon usage pattern of tobacco according to the Codon Usage Tabulated from GenBank (CUTG) website (http://www.kazusa.or.jp/codon/) [29]

**2.1.2 Prediction of solubility and stability of the recombinant protein, and its post-translational and allergenicity evaluation.** Before expressing this gene construct in tobacco plants, the final gene construct was translated *in silico* into a peptide sequence. SOLpro server (http://scratch.proteomics.ics.uci.edu/explanation.html#SOLpro) was used for predicting the solubility of the recombinant protein, and ProtParam online server (https://web.expasy.org/protparam/) for predicting various physicochemical parameters including amino acid composition, pI, aliphatic index (AI), instability index (II), *in vivo* and *in vitro* half-life, molecular weight (MW), and grand average of hydropathicity (GRAVY). The allergenicity of this recombinant protein was assessed by the AlgPred web server, which showed the post-translational modifications of CCL21/IL1β. NetOGlyc 4.0 Server and NetNGlyc 1.0 Server were used to identify the O-glycosylation and N-glycosylation sites of the recombinant protein, respectively; and NetPhos 3.1 Server was used to find the phosphorylation sites of the protein [29].

**2.1.3 Prediction of MHC I and II binding affinities of the recombinant protein.** T-cell and B-cell epitopes of the recombinant protein were identified utilizing BepiPred 2.0 (https://services.healthtech.dtu.dk/service.php?BepiPred-2.0), BCpreds (http://ailab-projects1.ist.psu.edu:8080/bcpred/), ABCpreds (http://crdd.osdd.net/raghava/abcpred/), SVMTrip (http://sysbio.unl.edu/SVMTriP/), and MAPPP online servers, (http://mendel.stanford.edu/Sidow

**Table 1. Prediction of T-cell epitopes of CCL21/IL1β protein.** A covering score over 90% and IC50 below 50 were used for selecting the best binder epitopes of MHCI and MHCII-related HLAs.

| MHCI | | | MHCII | | |
|---|---|---|---|---|---|
| Epitope peptide | HLA allel | score | Epitope peptide | HLA allel | IC50 |
| IPAKVVRSY | HLA-B*35:01 | 99.4% | LWVQQLMQH | HLA-DRB4*01:01 | 15.60 |
| LPRKRSQAEL | HLA-B*07:02 | 95.7% | ELWVQQLMQ | HLA-DRB4*01:01 | 17.10 |
| LCADPKELW | HLA-B*58:01 | 95.4% | PKELWVQQ | HLA-DRB4*01:01 | 18.50 |
| LCADPKELW | HLA-B*57:01 | 93.4% | AKVVRSYRK | HLA-DRB4*01:01 | 19.20 |
| IPAKVVRSY | HLA-B*53:01 | 90.3% | QQLMQHLDK | HLA-DRB4*01:01 | 21.40 |

Lab/ downloads/*MAPP*/index.html) (Table 1). These bioinformatic tools have been used to predict antigenic epitopes present on T-cell and B-cell surfaces by primary histocompatibility complex class I and II molecules (MHC I, MHCII) [30]. In addition, the TAPPred server was used to verify the binding affinity of the recombinant protein toward TAP transporter. This online service is based on cascade SVM and uses the sequence and properties of amino acids.

**2.1.4 Homology modeling structure using *in silico* tools.** Since the three-dimensional (3D) structure of the recombinant protein is not available, comparative modeling method was initially used to create the 3D structure of the recombinant protein. Comparative modeling is one of the best approaches to prediction of the 3D structure of the target protein, in which 3D structures with sequences very similar to the target sequence are used as templates. One of the best software for comparative modeling is MODELLER. The authors used Modeller 9.24 for 3D modeling of the target. Ten models were produced, and the one with the highest interaction and connectivity between the aforementioned epitopes was selected as the best model [31].

**2.1.5 Molecular dynamics simulation and prediction of the stability and flexibility of the recombinant protein.** Molecular dynamics simulation was performed using Gromacs 2019.6 software. Input structures were prepared with ff99SB force field. The correct hydrogen bonding patterns of histidines was defined for all proteins. The surface charge of the structure was neutralized by adding chlorine ions. The protein was placed in a layer of 8 angstrom-thick TIP3P water molecules inside an octahedron box using gmx solvate software. Reduction of energy on the structures was achieved with 50,000 steps by the steepest descent method to eliminate van der Waals interactions and hydrogen bonds between water molecules and the complex. In the next step, temperature of the system was gradually increased from 0 to 310 K for 200 ps in constant volume, and then the system was equilibrated at constant pressure for 200 pc. Molecular dynamics simulations were performed at 37˚C for 100 nanoseconds. Non-bonded interactions with 10-angstrom intervals were calculated by the PME method. The SHAKE method was used to restrict the hydrogen atoms involved in bonding in order to enhance computing performance. Finally, simulation data were stored at 0.4 pc intervals for analysis [28]. Two of the best methods for such analyses are root mean square deviation (RMSD) and Radius of Gyration (RG) change over time. The RMSD between the structures created during molecular dynamics simulation in the time dimension is a suitable and common standard to ensure the structural stability of the protein. Therefore, RMSD changes related to alpha carbon atoms of the protein during simulation time (100 nm) relative to the original structure were calculated and extracted. RG is another crucial parameter used in studying changes of protein size during simulation of molecular dynamics.

**2.1.6 Docking and molecular dynamics simulation of the CCL21/IL1β and CCR7 receptor.** This project aimed to investigate the interaction between chemokine ligands and CCR7 receptors. The CCR7 protein is intermembrane and, therefore, to simulate the CCR7-CCL21 complex, this complex must first be located inside the plasma membrane. For this purpose, the

CHARMM-GUI server (https://www.charmm-gui.org/) was used to place the complex inside the plasma membrane. POPC phospholipid was selected to generate the lipid membrane. After placing the protein complex inside the POPC membrane according to the protocol in the CHARMM-GUI server, the desired output was selected for molecular dynamics simulation with GROMACS 2019.6 software. At this stage, after energy minimization using the steepest descent algorithm, the system was balanced in NVT conditions in a time step of 1 femtosecond for 1 nanosecond. Equilibration was then performed under NPT conditions in a time step of 2 femtoseconds for 4 nanoseconds. The Berendsen algorithm was used to keep the temperature constant at 310 K and the pressure at 1 atmosphere in NVT and NPT conditions. Also, during the equilibration process, a series of restrictions according to the protocols available on the CHARMM-GUI server was used to limit proteins, water, and phospholipid molecules. After the equilibration steps were completed, molecular dynamics simulations were performed in the production phase for 100 nanoseconds. At production stage, Nose-Hoover and Parrinello-Rahman algorithms were used to keep the temperature and the pressure constant, respectively. The simulation time step at production stage was 2 femtoseconds, and all the limitations of solvent and soluble molecules were removed at this stage. There were 179 POPC(phosphatidyl-choline$^)$ and 20443 water molecules, and 54 and 89 sodium and chlorine atoms, respectively. The simulation time was 100 nanoseconds. The cut-off values for electrostatic and van der Waals interactions were 1.2 and 1 nm, respectively. RMSD and RG were used to study the stability of the vaccine on CCR7 receptors during molecular dynamics simulation.

## 2.2 Stable transformation of tobacco plants with the *ccl21/IL1β* gene

Sterilized tobacco (*Nicotiana tabacum* L) seeds were germinated on MS medium with 15 g/L sucrose, solidified by 7g/L agar. The recombinant plasmid (pBI121-*ccl21/ IL1β*) was introduced into *Agrobacterium tumefaciens* strain GV3101 via the freeze and thaw method [32]. Afterward, 200μl of *Agrobacterium* suspension, containing recombinant plasmid pBI121, was transferred to 50 ml of fluid medium containing kanamycin (50 mg/l), rifampicin (50 mg/l) and gentamicin (20 mg/l), and incubated for 24 hours at 28°C. The culture was centrifuged at 5000 rpm for 5 min at 4°C, and the cells were resuspended in inoculum medium consisting of three parts MS medium (pH 5.2) and one-part LB medium with 0.05 mM acetosyringone (3′, 5′-dimethoxy- 4′-hydroxy acetophenone). The suspension was placed in a dark room incubator at 28°C for half an hour. Sterile leaf disks (about 0.5 cm$^2$) were excised and inoculated with recombinant *Agrobacterium tumefasciens [33]*. Explants were transferred to a co-culture medium (MS with 0.2 mg/L BA and 0.1 mg/L NAA without antibiotics) and cultured for 48 hours in a dark room incubator at 25°C. After co-cultivation, the explants were transferred to the branching induction medium (MS containing 30g/L sucrose, 1 mg/L BAP, 0.1 mg/L NAA, and 300 mg/L cefotaxime and 50 mg/L Kanamycin antibiotics). The selection process was continued for 14 days at 22°C on a photoperiod cycle of 16 hours light and 8 hours dark (LD 16:8). The selected explants were then transferred to solidified root-inducing medium (MS basal salts with 15 g/L sucrose, 0.2 mg/L NAA, and 200 mg/L cefotaxime). Finally, the plants were transferred to the greenhouse with a temperature of 23 to 25° C under light regime, 16 hours of light and 8 hours of darkness. until the seeds of the $T_1$ generation were obtained. Growth and self-sufficiency of $T_1$ plants were analyzed to confirm transgene stability [34].

## 2.3 PCR and real-time PCR assays

Genomic DNA of transgenic plants was identified by PCR analysis using two primers, *CaMV-ccl21* F 5′ GATGACGCACAATCCCACT 3′ and *CaMV-ccl21* R 5′ CCCTTTCCCTTCTTT

CCA3′, which amplify 386 bp around the attachment region between CaMV promoter and the *ccl21* gene. In addition, *nptII* F 5′ CACGGTTCAACAACATCCAG3′ and *nptII* R 5′ TGAA-GACCCTGACTGGGAAG3, amplifying 782 bp around the Kan R region in PBI121 plasmid were used. PCR was performed according to the temperature profile provided in the appendix: 94ºC 1 min, 50ºC 1 min, and 72 ºC for 2 min for 30 cycles (Parstous kit; Cat.No C101081).

Whole RNA was extracted from infiltrated leaf tissue (Parstous kit; Cat. No A101231), and complementary DNA (cDNA) was synthesized through reverse transcription using oligo[23] primer (Parstous kit; Cat.No C101131). The resulting cDNA mixtures were utilized as templates for real-time PCR. Real-time PCR (BioRad) was performed in a 20 µL reaction volume containing 1 µM of each primer and 10 µL of SYBR green real-time PCR master mix (Parstous kit; Cat.No C101021). Real-time PCR experiments were carried out for each sample in duplicate. Forward and reverse primers for real-time PCR were 5′ GGGTTCAACAACTTATGC 3′ and 5′ CTTTCCCTTCTTTCCAGT 3′, respectively.

## 2.4 Extraction and purification of total soluble protein from T2 plants

Total Soluble Protein (TSP) was extracted using 500 µL of 1x PBS as a biological buffer and 10 µL of 100mM phenyl methane sulfonyl fluoride (PMSF) per 0.2 g leaf dry mass. The extract was centrifuged at 12,000 g for 20 min at 4ºC. The extracted TSP was purified by Ni-IDA resin (Cat No: A101271), and also dialysis was used to remove the imidazole contained in the protein purification buffer. Purified CCL21/IL1β concentration was determined as % of TSP and µg/g of leaf Fresh Weight (FW).

## 2.5 Protein dot blot assay and western blot assay

Purified recombinant protein in transgenic lines was measured by standard protein dot blot assay. First, 10µl of TSP from transgenic lines was dotted onto the nitrocellulose membrane, and the membrane was left to dry. Next, the membrane was incubated with bovine serum albumin (BSA) as the blocking solution for 1 hour. After incubation, the membrane was thoroughly washed three times with PBST/PBS and incubated with conjugated anti-6x His tag® mouse monoclonal antibody for 1 hour at 37ºC, then washed three times with PBST/PBS, and finally incubated with DAB (diaminobenzidine) substrate. A small amount of commercial CCL21 antigen containing Histidine tag (1 µL) was used as the positive control (R&D systems; Cat No: 966-6C), and 10µl of protein from the wild-type plant was used as a negative control. For western blotting, 100 micrograms of TSP per sample was resolved on 12% SDS-PAGE and visualized by Coomassie Brilliant Blue staining. For western blotting, the resulting SDS-PAGE was transferred onto a PVDF membrane by electroblotting (Biometra) for 2 hours at room temperature at 120 V. The membrane was then blocked with blocking buffer [BSA] PBS] for 1 hour at room temperature or overnight at 4ºC. Next, the membrane was cleaned three times with washing buffer [PBST]—each time 10 minutes at room temperature, then probed with the conjugated anti-6x His tag® mouse monoclonal antibody (Sigma-Aldrich) overnight at 4ºC or 1 hour at room temperature and rewashed three times with washing buffer (CMT; Cat. No CMGWBT). Finally, the protein bands were visualized by staining the membrane with TMB substrate (Promega; Cat.NO W4121).

## 2.6 Enzyme-linked immunosorbent assay (ELISA)

ELISA plates were coated with total soluble proteins from the wild type and transgenic plants and CCL21 antigen at 37˚C for 1 hour, and then incubated with 1% BSA w/v in PBS for 2 hours at a temperature of 37˚C. The wells were washed using PBST/PBS and incubated with the antiserum (conjugated anti-6x His tag® mouse monoclonal antibody), which was

reactivated against CCL21 protein (1:1000 dilutions) [35]. Wells were developed with TMB substrate. The color reaction was stopped by $H_2SO_4$ (2N concentration), and absorbance was read at 405 nm wavelength (CMT; Cat.No CMGTMB-100).

## 2.7 FTIR spectrometry

The sample concentration in KBr should be in the range of approximately 0.2% to 1% [36]. To prepare a KBr tablet, about 8.1 solids and 0.25–0.50 teaspoonful of KBr were thoroughly mixed in a mortar while grinding with the pestle. The material was pressed at 5000–10000 psi by pellet press instrument (following pellet press brochure instructions). The compressed sample was carefully removed from the pellet and placed in the Fourier-transform infrared (FTIR) spectrometer's sample holder. Spectral analysis of UV absorption at 400–4000 $cm^{-1}$ was performed by UNICAM UV 100 UV/visible light spectrophotometer. FTIR spectra were recorded on a Thermo Nikolet AVATAR370 spectrometer.

## 2.8 MALDI TOF/TOF mass spectroscopy

The purified recombinant protein was analyzed for purity by SDS-PAGE stained with G-250 Coomassie Blue. Protein spots were excised from preparative stained gels, and Gel slices were destained with wash solution [100% acetonitrile and 50 mM ammonium bicarbonate $(NH_4CHO_3)$] for 1 hour at room temperature. The protein spot was then air-dried for 30 min at 37˚C. Protein was digested using a trypsin solution (12 ng/ml trypsin in 50 mM $NH_4CHO_3$) by incubation for 45 min at 47˚C. Excess of trypsin solution was removed and replaced by 50 mM $NH_4CHO_3$, and the gel slice was incubated overnight at 37˚C [37]. Samples were sent to the Centre of Mass Spectrometry of York University to analyze the plant synthetic protein by conventional ionization methods.

## 2.9 *In Vitro* wound healing assay

There are five phases for wound healing assay; hemostasis, inflammation, cellular migration and proliferation, protein synthesis and wound contraction, and remodeling. However, mainly only three phases, inflammatory, proliferative, and remodeling are presented due to the overlap of phases. So potential activity of purified CCL21/IL1β in stimulating the proliferation and migration of MCF7CCR7+ cancer. *In vitro* wounds healing assay were induced using a modified protocol adapted from Phan *et al.* (2001). Clinical studies have revealed that MCF7 breast cancer tumors do express CCR7 [38]. In order to elucidate the functional role of CCL21/CCR7, MCF7 cells obtained from Razavi Khorasan branch of Academic Center for Education Culture and Research (ACECR) (Mashhad, Iran) were seeded in 6-well plates at densities of 6×150 cells/well in the DMEM high glucose growth medium. After reaching 95% confluency, monolayers of cells were scratched using a sterile pipette tip and the cells drawn firmly across the dish PBS was used for washing the cells to remove the loosened debris/ for washing out non-adherent cells. 7.5 µg/ml of recombinant plant protein, commercial protein (as positive control), and non-recombinant protein (as negative control) were added to each series of wells. Moreover, the cells were treated with an equivalent amount of DMSO as control. For migration assay, cultures were rinsed twice using PBS solution, fixed by absolute methanol, stained by Giemsa, and inspected by a light microscope equipped with a calibrated ocular lens at 40x magnification. Images were recorded 24, 48, and 72 hours after wounding. Cell migration rates were quantified by evaluating the changes in the wound area (pixels) using Image J 1.46 r software.

## 2.10 MTT cytotoxicity assay

MTT assay was used to determine the half-maximal inhibitory concentration ($IC_{50}$) of all recombinant protein concentrations in this study. series of dose-response data (e.g. protein concentrations $x_1$, $x_2$,. . ., $x_n$ and viability percentages $y_1$, $y_2$,. . ., $y_n$) were needed for calculating IC50. Y values were ranging from 0 to 1. Therefore, the most precise method for estimating $IC_{50}$ is x-y plotting, followed by fitting the corresponding data with a straight line (linear regression). The $IC_{50}$ value was then estimated using the fitted line [39].

$$Y = a * X + b$$

$$IC50 = (0.5 - b)/a.$$

After being seeded at a density of $8 \times 10^3$ cells/well in 96-well plates for 24 hours, the cells were treated with different concentrations of CCL21 recombinant protein, commercial CCL21, and non-recombinant protein (2.5, 5, 7.5, 10 μg/ml for 24, 48, and 72h). Next, 5 mg/ml MTT solution (Sigma, Germany) was added to each well. Plates were incubated at 37˚C for 4 hours. Finally, the insoluble formazan was dissolved in DMSO. Optical density of each well was measured spectrophotometrically at 570 nm (all tests were performed in triplicate). The viability of cells in each treatment was calculated by dividing the absorbance of treated cells in each concentration to the mean absorbance of control cells.

## 2.11 Under agarose gel assay

Under agarose gel assay was used to investigate the mechanism of CCl21 impact on T-cell migration because chemokines are cytokines involved in leukocyte migration; Chemotaxis is the primary mechanism by which cell movements are directed within multicellular organisms, and it is a major component of embryonic development, wound healing, and immune responses. Chemotaxis involves a complex cascade of events formation of signaling complexes, receptor polarization, adhesion molecule activation, and cytoskeletal reorganization [40].

**2.11.1 PBMC isolation from whole blood.** The Ficoll method was used for separating and isolating PBMC (peripheral blood mononuclear cells contain CD4[+] and CD8[+] and CCR7 Expression in Human Primary PBMCs [41]). To isolate PBMC from whole blood, 10ml Ficoll was added to 4ml blood in a 50 ml tube, and then the blood was diluted with 2X phosphate-buffered saline (2X PBS) and 2% fetal bovine serum albumin (2% FBS). Finally, 5 ml of PBS solution was added. Diluted blood rose to the top of Ficoll solution. The tube was centrifuged at room temperature (15–25˚C) for 30 minutes at 400 g, with the brake in the off position. The upper plasma layers, where the lymphocytes were located, were carefully removed, and the mononuclear/lymphocyte cell layer at the plasma-Ficoll interface was transferred to a clean tube. At least 3x the volume of balanced salt solution was added to the mononuclear/lymphocyte cells in the test tube. The cells were centrifuged at 400 g for 10 minutes at room temperature [42].

**2.11.2 Agarose gel well formation for chemotaxis assay.** 1 percent agarose solution was prepared in a mixture containing 50% DMEM, 10% FBS, 50% PBS, and 2 mM L-glutamine to form wells on the agarose gel. 1.6 mL of the 1% agarose solution was added to each 10-mm sterile petri dish. To humidify the gel, 5 mL DMEM was added to each petri dish after allowing the gel to cool down for 20 min. Afterward, 5 mL FBS-free DMEM was added to the gel (1–6 hours before performing the cell migration assay) and three small wells with a distance of 10 mm were cut in culture medium/ in each petri dish [40, 43].

**2.11.3 Chemotaxis assay and measurements during cell migration.** PBMC cells were seeded at a density of $1 \times 10^5$ cells in the middle well with 10% FBS-DMEM. After 24 hours,

the medium was replaced by FBS-free DMEM [40]. Chemokines (commercial CCL21, recombinant protein) were then placed in one of the neighboring wells to exert their chemoattractant effect, with FBS-free DMEM or non-recombinant protein in other wells as negative controls. Image capture and measurements were performed using an inverted microscope (Olympus).

**2.11.4 Cell processing and imaging; DAPI staining.** For DAPI staining, cell pellets were fixed in 4% paraformaldehyde (PFA) for 8 minutes (Sigma, Germany). Then, each well was washed 3 times (5 minutes for each wash) with 60 μL of 1x PBS for washing the cells. Afterward, the cells were permeabilized with 60μL/well of 0.1% Triton X-100 for 10 minutes and stained with 50μL/well of DAPI (1:2000 dilution, in 1x PBST) (Merck, Germany) for 10 min [40]. The stained cells were then counted using a fluorescent microscope. Chromatin condensation and nuclear fragmentation were the criteria to confirm apoptosis.

## 2.12 Statistical analysis

The data was classified into two categories: recombinant and non-recombined. In each category, there are four independent groups with concentrations of 2.5, 5, 7.5 and 10μg/ml and a control group. In each group, the experiment was repeated three times. In addition, the mentioned structure has been implemented in three periods of 24, 48 and 72 hours. By performing the Kolmogorov-Smirinov test on the obtained data, we conclude that the data has normal characteristics. In each time intervals, the concentrations of 2.5, 5, 7.5 and 10 will be compared with the control group, by using one-way analysis of variance and Scuffle's within-group test for recombinant and non-recombinant categories. Also, the paired T-test was used to compare recombinant and non-recombined groups. These tests are done by using SPSS 26.

## 3. Results

### 3.1 *In silico* analysis

**3.1.1 Recombinant construct properties.** The final recombinant construct was codon-optimized by GenScript Rare Codon Analysis Tool (https://www.genscript.com/tools/rare-codon-analysis), Codon usage plays a crucial role when recombinant proteins are expressed in different organisms. Therefore, to enhance efficient gene expression it is of great importance to identify rare codons in any given DNA sequence and subsequently mutate these to codons which are more frequently used in the expression host. Rare Codon Analysis Tool is powerful for codon usage frequency of your sequence and codon usage distribution. Codon Adaptation Index [44] was used for assessing the degree of codon optimization of this construct and for increasing the protein expression level; A CAI (Codon Adaptation Index) of 1.0 is considered ideal while a CAI of >0.8 is rated as good for expression in the desired expression organism So CAI = 1 was selected to support tobacco codon adaptation. A GC content of 48% was accepted as an appropriate expression (Fig 1A). The ideal percentage range of GC content is between 30% and 70%. Any peaks outside of this range was adversely affect transcriptional and translational efficiency. So OptimumGene can give you the option to solve this problem. The percentage of low frequency (<30%) codons based on your target host organism is 12%. This un-optimized gene employs tandem rare codons that can reduce the efficiency of translation or even disengage the translational machinery. Afterward, the synthetic *ccl21/IL1β* construct was cloned into the binary vector pBI121 [45]. Molecular mass (Mw), isoelectric point (pI) and solubility (%) of the recombinant protein were estimated to be 14,884.5504 Da, 9.377, and 0.8434, respectively by SOLpro server (http://scratch.proteomics.ics.uci.edu/explanation.html#SOLpro). The scaled solubility value is the predicted solubility, the population average for the experimental dataset (PopAvrSol) is 0.45, and therefore any scaled solubility value greater than 0.45 is predicted to have a higher solubility than the average soluble E.coli protein from the

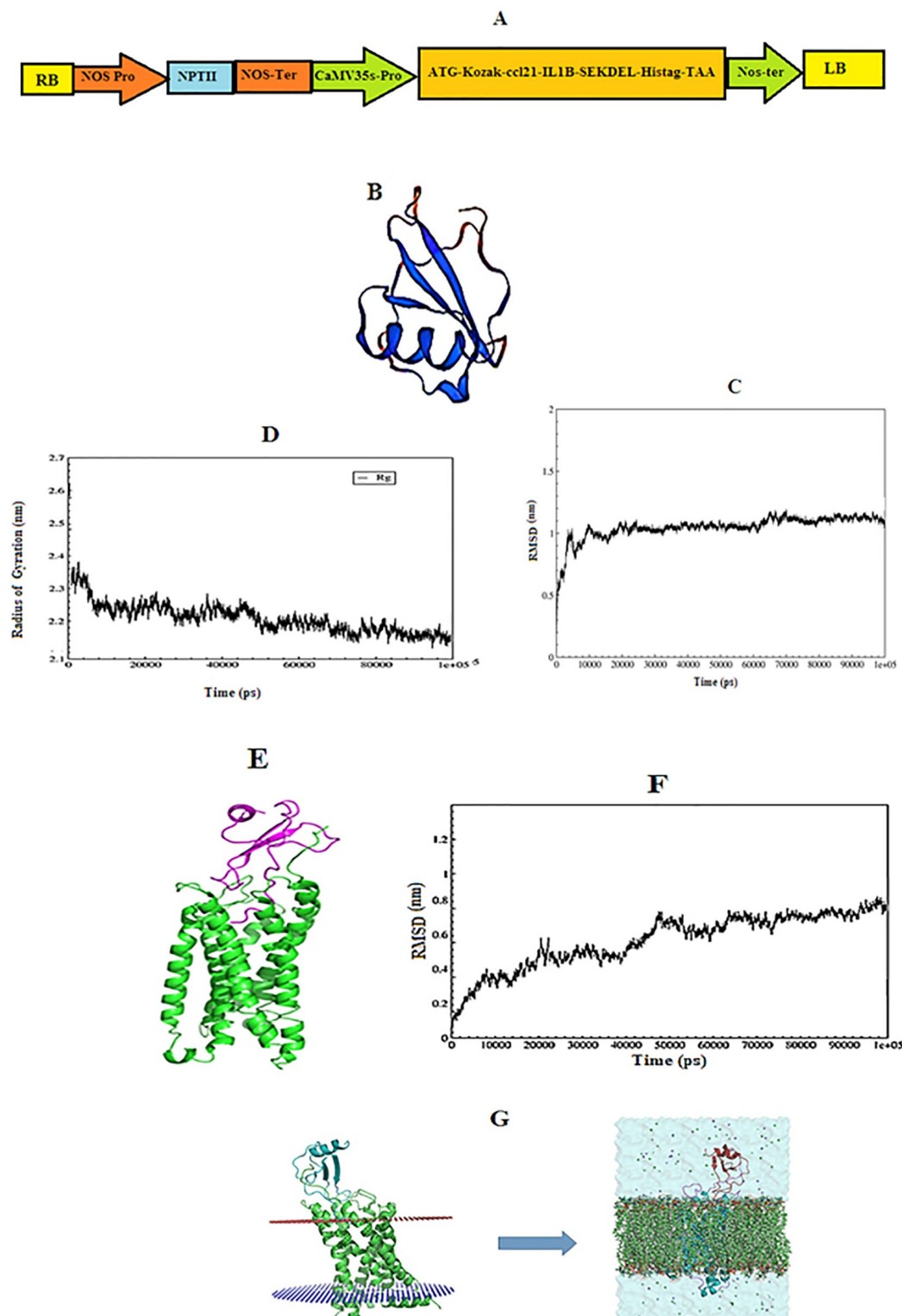

**Fig 1. *In silico* analysis of the recombinant protein. A**: Schematic representation of the pBI121-*ccl21/IL1β* construct. **B**: The best model for the recombinant protein as designed by Modeler 9.24. **C**: RMSD values in MD simulation of the recombinant protein: RMSD slope increased rapidly in the first 10 ns of MD simulation. RMSD reached 1 nm after about 10 ns, and 1.25 nm at 60 ns. Afterward, RMSD decreased slightly, reached 1.1 nm at 70 ns, and remained constant until the end of simulation. **D**: RG values in MD simulation of the recombinant protein: protein's RG was about 2.3 nm, but it

decreases rapidly in the following simulation, reaching 2.2 at about 10 ns, after which the slope of the radius decreased, and RG reached approximately 2.14 nm at about 80 ns. RG then remained constant until the end of the simulation, indicating the stability of the protein. **E**: Molecular docking image of the recombinant protein and CCR7 receptor. **F**: RMSD values in MD simulation of CCR7 and recombinant protein complex; RMSD increased at the beginning of the simulation. After 100 ns, RMSD reached 0.7 nm, after which it remained constant until the end of simulation. **G**: Placement of CCR7-CCL21/ IL1β complex inside the membrane by CHARMM-GUI server. This complex must first be located inside the plasma membrane. the number of phospholipids used in the membrane structure was 162 molecules of phosphatidylcholine.

experimental solubility dataset [46], and any protein with a lower scaled solubility value is predicted to be less soluble. The other physicochemical property of this recombinant protein was calculated by ProtParam tool (https://web.expasy.org/protparam/); The instability index is a measure of proteins, used to determine whether it will be stable in a test tube. If the index is less than 40, then it is probably stable in the test tube and If it is greater than it is probably not stable. So, the Instability index of this recombinant protein was calculated 38.03, which showed that it will be stable. The computed half-lives in mammalian, yeast, and *E. coli* cells were more than 30 h, 20 h, and 10 h, respectively. Aliphatic index and GRAVY were predicted to be 49.17 and -1.242, respectively. So high aliphatic index indicates that this recombinant protein is thermo-stable over a wide temperature range and GRAVY value for recombinant protein was calculated as the sum of hydropathy values of all of the amino acids divided by the number of residues in the sequence. The allergenicity of the recombinant protein was assessed, and accuracy of the prediction was 94% at the -0.4 threshold. Therefore, it was deemed not to be allergenic.

**3.1.2 Identification and evaluation of suitable epitopes.** Protein sequences encoding human CCL21 used in this structure have a DCCL motif, a domain binding to the CCR7 receptor, and a Pan HLA DR-binding epitope (PADRE) peptide sequence covering more than 90% of the HLA alleles. Since their IC50 is less than 50, they can bind to major histocompatibility complex II and I (MHCII/MHCI) molecules. In addition, the VQGEESNDK sequence of human IL-1β is involved in immune responses and has high adjuvant effects. These immunogenic regions of the recombinant protein were thus designed based on T helper lymphocytes [47], cytotoxic T lymphocyte (CTL) epitopes, and MHCI and MHCII. The aforementioned sequence constituted the principal part of the expression cassette. This recombinant protein binds to CCR7 receptors to form a complex which can bind to MHCI and MHCII molecules on T-cells and produce anti-metastatic and cytotoxic effects on cancer cell line chemotactic response in lymphocyte cells. Predicted epitopes of CTLs and THLs were selected based on the percentage of HLA allele coverage (Table 1). Epitopes with HLA coverage above 90% and IC50 below 50 were selected as the best epitopes for binding to MHCI and MHCII molecules. In addition, the BepiPred-2.0 predictions server showed that most of the recombinant protein sequences had a B-cell epitope position with a 0.5 threshold.

**3.1.3 Molecular modeling and dynamics simulation of the recombinant protein.** In this project, MD simulations were used for 100 ns to compare the conformational changes of recombinant proteins. Multiple measurements were analyzed throughout the simulation project, chiefly the root mean square deviation (RMSD) and the radius of gyration of the proteins with the time-dependent function of MD/ with MD as the time-dependent function. The results of these calculations for each simulation are shown in Fig 1C. The slope of RMSD increased rapidly in the first 10 ns of MD simulation. After approximately 10000 ps, RMSD reached 1 nm, which means that RMSD is equal to 1.25 nm at 60000 ps. Afterward, RMSD decreased slightly, reached 1.1 nm at 70000 ps, and remained constant until the end of the simulation. These results indicated the stability of the protein structure in this simulation (Fig 1C). Radius of gyration is an indicator of protein structure compactness. It is a concern with

how stable secondary structures are compactly packed into the 3D structure of the protein. The lowest radii of gyration and, accordingly, the tightest packing levels/ ratios are characteristic of α/β-proteins. As depicted in Fig 1D, gyration radius of the protein was approximately 2.3 nm at the beginning of the MD simulation, but it decreased rapidly in the next simulation and reached 2.2 at about 10000 ps. The slope of the radius decreased afterward until it reached 2.14 nm at 80000 ps and then remained constant until the end of the simulation, indicating the stability of the protein. In general, the Rg value indicates the compactness of proteins which in turn reflects their stability. The more Rg fluctuates, the less sable is the protein. Hence, it plays a significant role in comparative studies.

**3.1.4 Molecular docking and simulation of multi-epitope protein with the CCR7 receptor.** As mentioned, the purpose of this project was to investigate the interactions between ligands and CCR7 receptors. Molecular docking was performed and the best cluster had a score of -27, with a size of 34 complexes. In addition, Z-score was equal to -2.3. The image of this complex is shown in Fig 1E. The RMSD parameter was used to study the stability of the recombinant protein that bonded to the CCR7 receptor in molecular dynamics simulations and using the RMSD parameter. The corresponding RMSD diagram and the final structure are shown in Fig 1E and 1F. Fig 1E depicts the final structure of the MD complex, and it should be noted that water and membrane molecules have been removed in the image to show the protein structure more clearly. Fig 1F also represents the changes in the RMSD of the docked proteins over the simulation time. As shown in this figure, RMSD increased sharply to 0.4 nm by 10,000 ps and then it decreased and reached 0.3 nm at 15,000 ps, followed by an abrupt increase to 0.6 nm at 20,000 ps. A decrease and increase are observed in the RMSD diagram once again as the simulation continues. At 50,000 ps, the diagram reaches 0.7 nm and remains constant at the same value until the end of simulation, indicating the stabilization of the CCL21/IL1β protein when attached to CCR7 at 50,000 ps during the simulation.

The CCR7 protein is intermembrane and therefore, the CCR7-CCL21/IL1β complex must be simulated inside the plasma membrane. 162 phosphatidylcholine molecules were used for this purpose and formed the membrane structure. Also, a salt concentration of 150 mM NaCl was used to simulate physiological conditions and neutralize the system. In order to keep the ion molality of all systems 47 sodium and 64 chlorine atoms were added, respectively. The size of the system created for the length of the system was 8.22268 and 12.94010 nm, respectively. The TIP3P water model was used as a solvent, and the number of water molecules in the system was 17815. Cut-off values for electrostatic and van der Waals interactions were 1.2 and 1 nm, respectively. The pme algorithm was used to calculate long-range electrostatic interactions. The shake algorithm was used to keep the length of the bonds constant (Fig 1G).

## 3.2 Expression of *ccl21/IL1β* in transgenic plants

The regenerated transgenic plants were generation of T0 transgenic plants (Fig 2A). Genomic DNA from the leaves of T0 transgenic plants was isolated for PCR analysis. Most of the transgenic tobacco plants had 386 bp PCR products (Fig 3A). The third generation of transgenic plants was studied to select homozygous lines with one copy of this synthetic gene. The phenotype of T0 transgenic plants was Kanamycin Resistant (Kan R), and the genotype was heterozygous. The plants grown from seeds harvested from self-pollinated T0 transgenic plants were named T1 generation (T1 transgenic plants) (Fig 2C). The phenotypes of T1 plants were either Kanamycin Resistant (Kan R) or Kanamycin sensitive (Kan r) (Fig 2A and 2B). The ratio of phenotype classes was Kan R: Kan r = 3:1. Of the nine first-generation plants (T0). Four plant in the second generation (T1) had a single copy of the synthetic gene and followed Mendelian segregation rules. Plants grown from seeds harvested from self-pollinated T1 transgenic plants

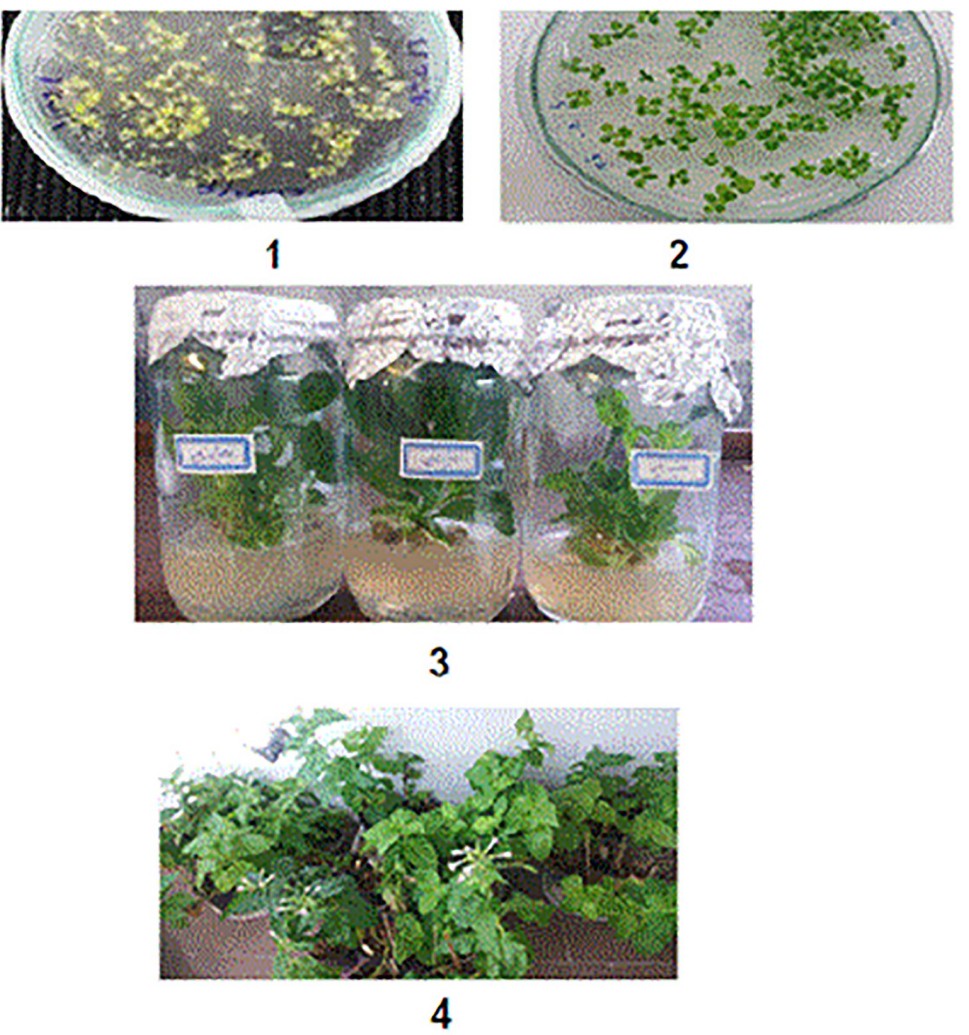

**Fig 2.** Stages of cultivation of T2 seeds in Kanamycin and growth of third generation transgenic plants (T2): 1) Seeds of second generation transgenic plants (T1) (2), Seeds of non-transgenic plants (3). Transgenic seedlings transferred to culture medium 4) Growing of transgenic tobacco plants (T2) in pots and starting the seed production stage for the fourth generation.

were named T2 generation (T2 transgenic plants). Of the T2 plants studied in the third generation (T2), the two lines (H2 and E1) did not differentiate in contrast to other lines regarding Kanamycin Resistance. Therefore, these two lines were considered to be third-generation homozygous lines. Moreover, the presence of synthetic construct and *nptII* in T2 plants was evaluated using PCR, and 782 bp bands were observed for DNA samples of transgenic lines (Fig 3B).

### 3.3 Evaluation of *ccl21/IL1β* expression by real-time PCR assay

The expression of *ccl21* was quantified using quantitative real-time PCR. Three samples from each transgenic line and one from the non-transgenic line (as control) were used for real-time PCR. Results indicated that the foreign gene was transcribed in the transgenic lines (Fig 3C). The results showed that H2 and E1 lines with high expression level of 2.97 times and 2.72

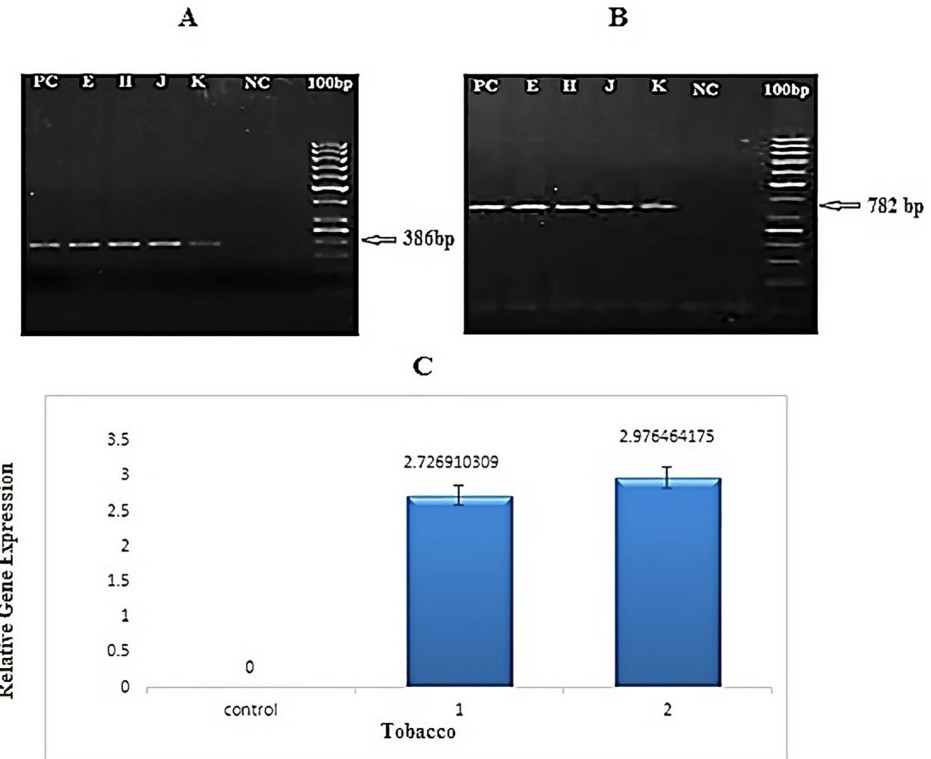

**Fig 3. PCR and real-time PCR analysis of *21/Il1β* gene in transgenic tobacco plants. A**: PCR product of *ccl21/Il1β* in T0 transgenic lines amplified by *CaMV-ccl21* primers (386bp). **B**: PCR product of *ccl21/Il1β* in T2 transgenic lines amplified by *nptII* primers (798 bp). (M: 100bp size marker. NC: PCR product of wild-type plant as negative control. H, J, K, and E: PCR products of transgenic plants. PC: PCR product of pBI121-*ccl21* construct as positive control). **C**: Quantitative measurement of *ccl21/Il1β1* transcription in transgenic lines of tobacco via real time-PCR. 1 and 2: two samples from transgenic lines; Control: negative control (non-transgenic plant). data showed that two lines H2 and E1 did not differentiate in contrast to other lines but results showed that H2 and E1 lines with high expression level of 2.97 times and 2.72 times higher than the control (non-transgenic line), respectively.

times higher than the control (non-transgenic line), respectively. Also, data showed that two lines H2 and E1 did not differentiate in contrast to other lines.

## 3.4 Detection of purified recombinant protein

The expected 55 kDa band was seen under reducing conditions on SDS-PAGE for the purified protein sample from the homozygous transgenic line, indicating expression of the recombinant protein in the transgenic lines (Fig 4A). So, results showed the expected bands at a molecular mass of 63 kDa in the denaturation condition of the SDS gel.Dot blot and western blot assays confirmed the expression of recombinant protein. In the dot blot assay, no signal was observed for wild-type plants, whereas expression of the gene of interest in transgenic lines was confirmed (Fig 4B). In western blot analysis, the recombinant protein with an estimated molecular mass of 55 kDa was successfully detected under reducing conditions (Fig 4C). So, results showed that; after purification, the recombinant protein did not migrate at the predicted molecular weight of 15.5 kDa under denaturing conditions, suggesting that the protein's mobility under denaturing conditions was influenced by its biochemical properties. So, the purified recombinant protein was eluted as a single peak corresponding to a molecular weight of 55 kDa, suggesting that the protein might exist in a dimeric state in solution. Also as shown

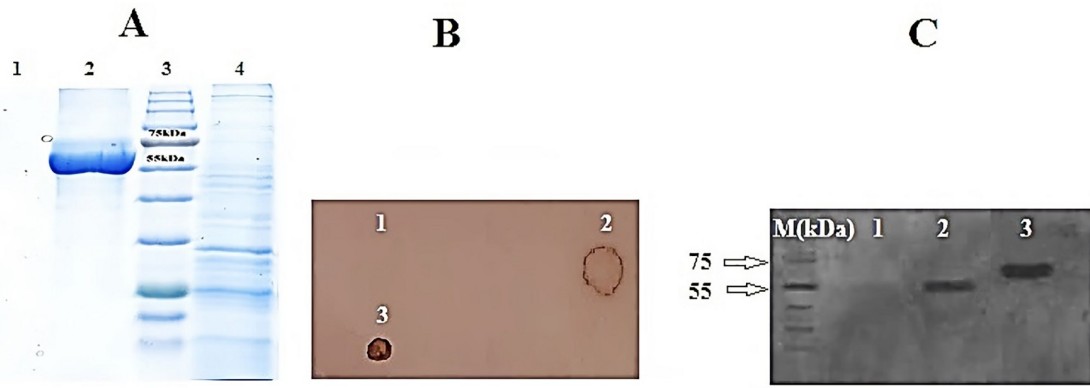

**Fig 4. Detection of purified recombinant protein. A**: SDS-PAGE of purified protein from transgenic lines and non-transgenic tobacco. Lane 1: non-transgenic tobacco as negative control; Lane 2: transgenic line; Lane 3: Fermentase Prestained Protein Ladder (SM1811); Lane 4: total soluble protein. **B**: Protein dot blot for detection of recombinant protein in transfected leaves of tomato. (1): positive control, (2, 3): protein sample of transfected plants and (4): protein sample of non-transfected plant. **C**: Western blot of CCL21 protein from transgenic tobacco leaves using conjugated anti-6x His tag® mouse monoclonal antibody (Sigma-Aldrich). A clear 55 kDa band was observed for homozygous lines and a 65 kDa band for commercial CCl21 antigens. M: Fermentase Prestained Protein Ladder (SM1811), 1: protein from non-transgenic plant as negative control; 2: protein sample from transgenic plant; 3: Commercial CCL21 antigen) (R&D systems; Cat No: 966-6C), (as positive control. After purification, the recombinant protein did not migrate at the predicted molecular weight of 15.5 kDa under denaturing conditions, suggesting that the protein's mobility under denaturing conditions was influenced by its biochemical properties. So the purified recombinant protein was eluted as a single peak corresponding to a molecular weight of 55 kDa, suggesting that the protein might exist in a dimeric state in solution. Also as shown in Recombinant Human CCL21/6Ckine as commercial CCL21 ((R&D systems; Cat No: 966-6C)), The predicted Molecular mass is 38.3 kDa but under reducing condition the molecular mass of commercial CCL21 was about 65 kDa.

in Recombinant Human CCL21/6Ckine as commercial CCL21 (Cat No: 966-6C), The predicted Molecular mass is 38.3 kDa but under reducing condition the molecular mass of commercial CCL21 was about 65 kDa.

Besides, ELISA assay was employed to quantify the expression of recombinant protein, and a standard curve was drawn by ELISA using concentrations of commercial CCL21 (Cat No: 966-6C) over the linear range from 0.25 to 2.5 μg/ml with 0.5 μg/ml increments. The experiment showed that the recombinant protein expression levels were 594 μg/g and 342 μg/g of fresh weight (FW) for two transgenic lines, and these lines were named H2 and E1 respectively. Total Soluble Protein (%TSP) of recombinant protein was measured to be 2.14% and 1.81% for H2 and E1 transgenic lines, respectively. In contrast, no significant differences were observed for non-transgenic plants. Results of ELISA analysis for the expressed recombinant protein from the transgenic lines were in agreement with those of CCL21/IL1β expression analysis conducted by a conjugated anti-His antibody.

## 3.5 Evolution/ Evaluation of CCL21 secondary structure in commercial, recombinant and non- transgenic protein by FTIR assessment

Secondary structures of commercial CCL21, purified CCL21 recombinant protein, and non-transgenic protein were analyzed by FTIR spectroscopy at 400–4000 cm$^{-1}$, as illustrated in Fig 5. The recombinant protein showed similar peak positions to the commercial antigen throughout most of the absorption range which indicates that both proteins had a similar molecular structure. However, there was a difference in the OH and NH$_2$ region (3000–3500 cm$^{-1}$). Furthermore, a peak at 3430 cm$^{-1}$ in recombinant protein and one at 3425 cm$^{-1}$ in the commercial antigen were observed. Another difference in the two samples was related to the absorption region of the CO amino acid (1500–2000 cm$^{-1}$). An absorption spectrum of 1645 was observed in the purified recombinant CCL21, while 1639 cm$^1$ was observed for the commercial antigen,

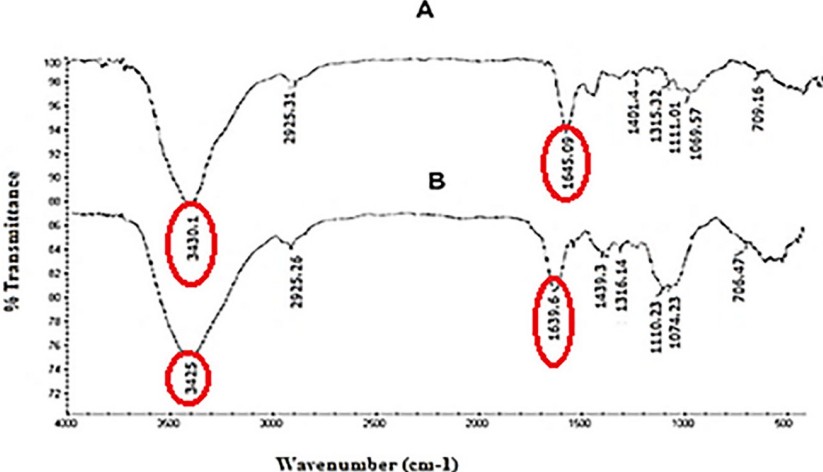

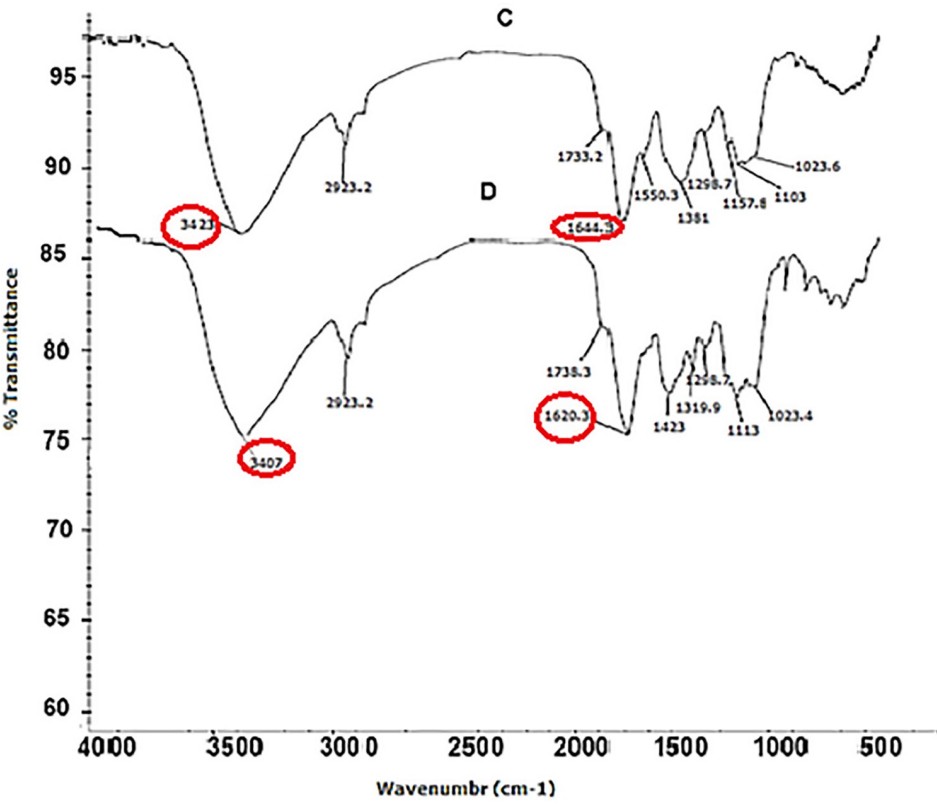

**Fig 5. Structure analysis of purified recombinant protein (A) and CCL21 antigen (B), transgenic line (C) and non-transgenic line (D) by FTIR.** Similar peak positions were observed in the spectra of A and B; differences can be observed in 3000–3500 $cm^{-1}$ region (the peak at 3430 $cm^{-1}$ in A and the peak at 3425 $cm^{-1}$ in B), which are related to OH and $NH_2$ in B, CO carbonyl (1500–2000 $cm^{-1}$) has an absorption spectrum of 1645 in purified A, while corresponding value is 1639 $cm^{-1}$ in B. The spectra showed different peak positions in C and D; the peak was at 3423 $cm^{-1}$ in C and 3407$cm^{-1}$ in D. The absorption spectrum was 1644 in C and 1620 $cm^{-1}$ in D.

which could be related to the presence of 9 different amino acids in the recombinant sample which were absent in the commercial antigen. The data revealed many disparities in the different absorption bands of the two plants, indicating that CCL21 has been expressed with 134 amino acids in the transgenic plants and, presumably, that the two samples had different molecular structures. The most important difference is related to the OH and $NH_2$ region (3000–3500 $cm^{-1}$). Moreover, a peak at 3423 $cm^{-1}$ in the transgenic line and at 3407 $cm^{-1}$ in non-transgenic tobacco was shown. Another difference was in the CO amino acid absorption region (1500–2000 $cm^{-1}$). Finally, absorption peaks of 1653 1620 $cm^{-1}$ were observed for the transgenic line and the non-transgenic tobacco plant, respectively, which is due to 134 different amino acids in transgenic samples and could be the main reason for the differences we observed between the transgenic and non-transgenic tobacco plants.

### 3.6 Analysis of MALDI-TOF/TOF mass spectra of the recombinant protein

MALDI-TOF/TOF results were analyzed using Mascot software. Searches used the concatenated Mascot Generic Format (mgf) file. Mascot software identified two peptides matching the expected CCL21/IL1β sequences with a high score. Mascot analysis demonstrated that CSIPAILFLPR and QGEESNDK sequences are part of human CCL21 and ILIβ with molecular weights of 1229.55 Da and 947.1 Da, respectively, whereas no tobacco sequences were detected. The average protein sequence coverage was 12%, which confirmed recombinant CCL21/IL1β as the target protein that was correctly expressed and purified from transgenic lines.

### 3.7 Migration of the cancer cell line into the scratch

Migration rate of MCF7 cells was estimated in three groups: a group with commercial CCL21 antigen as the positive control, the medium culture group without plant protein extract as the negative control, and the purified protein extract from transgenic tobacco leaves as the third group. Medium concentrations of the extract (7.5 μg/ml) induced migration of the cancer cells and closure of the scratch, and the percentage of closure was calculated using Image J. software. When assessed 24 hours post-treatment, the two control groups had migrated by 22% to scratch. After 48 hours, the scratch closure was 88%. Surprisingly, 72 hours following the same treatment, the scratch was almost completely closed. However, 24 hours after treatment with the same concentration of the purified protein extract of transgenic leaves, no evident migration was detected for the cells; but 15% and 28% scratch closure occurred after 48 and 72 hours, respectively. The results suggest that the extraction of plant recombinant protein was negatively related to the migration rate of the cells (Fig 6).

### 3.8 Cytotoxicity of CCL21/IL1β as recombinant protein

MCF-7 Human breast cancer cells [13] were treated with different concentrations of CCL21/IL1β recombinant protein, non-recombinant protein as the negative control, CCL21 commercial protein as the positive control, and also DMEM high glucose culture medium as the control, for 24, 48, and 72 hours to investigate the cytotoxicity of CCL21/IL1β using the MTT test. As summarized in Fig 6A to 6C; CCL21/IL1β induced its toxic effects in a dose- and time-dependent manner. 72 hours after treatment with a concentration of 7.5 μg/ml, survival rate of cancer cells incubated with the CCL21/IL1β purified recombinant protein was 28% compared to DMEM (Fig 7A). The $IC_{50}$ of CCL21/IL1β plant recombinant protein against MCF7 cells was 3.86 μg/ml. Interestingly, in comparison with this purified plant recombinant protein different concentrations of commercial CCL21 were used according to the standard curve of % TSP, where CCL21/IL1β antigen promoted its toxic effects at a concentration of 7.5 μg/ml, the

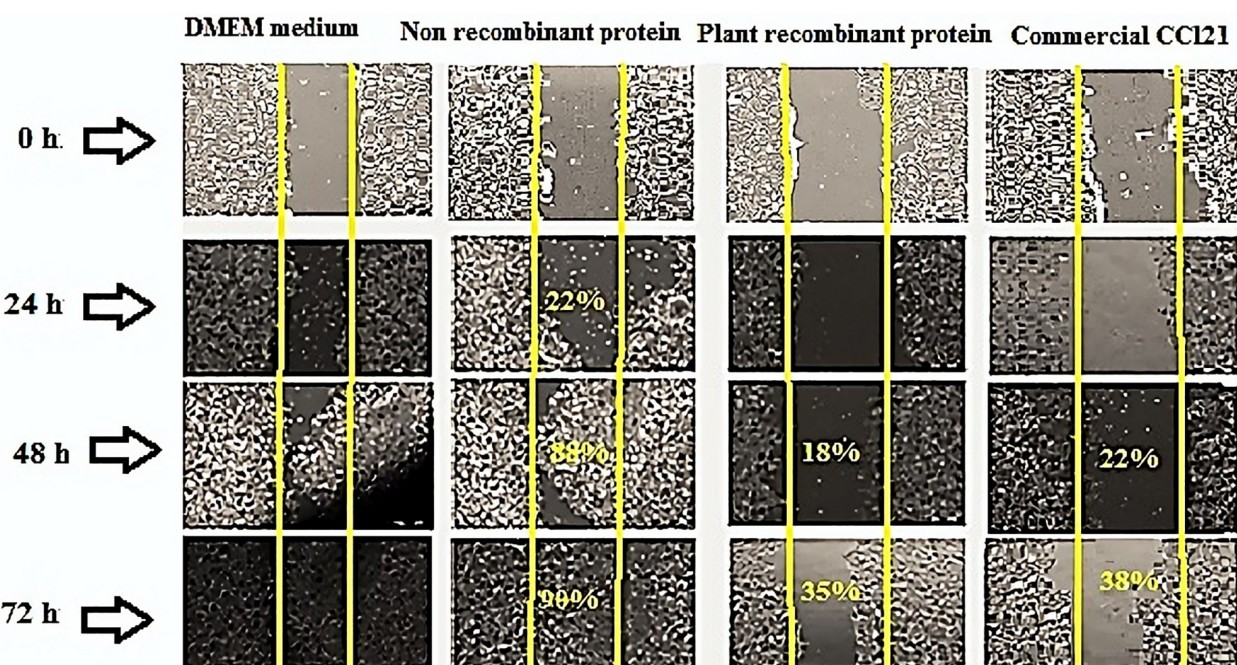

**Fig 6. Photomicrographs of *in vitro* wound assay.** The initial width of scratches in cancer cell cultures was estimated between 700 to 800 μm. MCF7 migration rates were assessed at 0, 24, 48, and 72 hours after wound induction in each group. Treatments were non-recombinant protein extract and DMEM culture medium as negative controls, plant recombinant protein, and commercial CCl21 as the positive control. Medium concentrations of the extract (7.5 μg/ml) induced migration of cancer cells and closure of the scratch, and closure percentage was calculated using Image J. software. When assessed 24 hours post-treatment, the two control groups had 22% migration to the scratch area. After 48 hours, 88% scratch closure was observed. Surprisingly, 72 hours following the same treatment, the scratch was almost completely closed. With the same concentration of the purified protein extract of transgenic leaves, however, no noticeable migration of the cells was detected 24 hours after treatment, and 15% and 28% scratch closure rates occurred after 48 and 72 hours, respectively.

survival rate of cancer cells was 22.1%, and IC$_{50}$ was 0.315 μg/ml (Fig 7B). Therefore, various concentrations of commercial CCL21 were used for drawing the standard curve to determine the percentage of TSP of the recombinant protein; and commercial CCL21 antigen at concentrations of 0.2, 0.5, 1, and 2.5 μg/ml were used to determine the maximum amount of antibody. At a concentration of 7.5 μg/ml, survival rate of cancer cells incubated with non-recombinant protein was 74.3% lower compared to DMEM as control (Fig 7C). In investigating the hypothesis of "Significant difference between the concentrations (7.5 with other concentrations) in the time period of 72 hours", we performed the ANOVA statistical test with Scheffe's within-group test for recombinant, non-recombinant and commercial proteins separately. Since the calculated P-Value for the comparisons of (2.5 and 7.5), (5 and 7.5) and (10 and 7.5) was less than 0.05, so the hypothesis is accepted.

### 3.9 Migration of PBMC cells (chemotaxis assay)

CCR7 cytokine is expressed in human primary PBMCs, to observe whether PBMC cells and chemoattractants attract each other, PBMC cells can be seeded in the middle well and chemoreactants in the wells to its right and left [43]. Four versions of simultaneous migration induction groups were used in this experiment. The first group was the simultaneous stimulation of commercial CCL21 as the positive control (2.5 μg/mL) in the left well and DMEM medium as the negative control (2.5 μg/mL) in the right one. The results confirmed that T-cells responded similarly to CCL21/IL1β stimulations. As a result, we obtained approximately 170 migrating cells after 24 hours of stimulation (Fig 8A). Migration kinetics were also very similar for

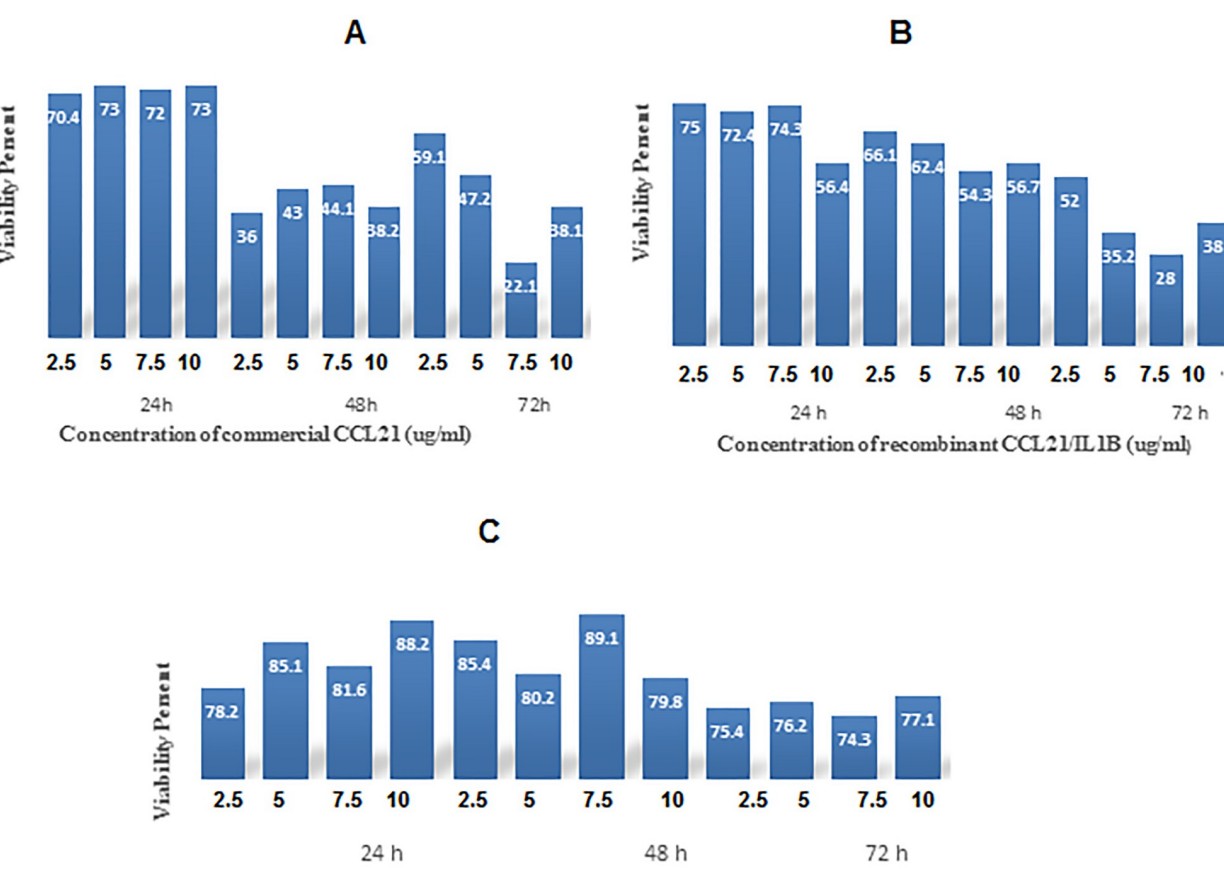

**Fig 7. Cytotoxicity analysis of recombinant, commercial, and non-recombinant CCL21 on MCF7 cell line at 24, 48, and 72 hours after incubation. A**: Cytotoxicity of recombinant CCL21 protein in comparison with the control group at different concentrations (2.5, 5, 7.5, and 10 μg/ml) on MCF7 cancer cells: at 7.5 μg/ml, the survival rate of cancer cells incubated with the purified recombinant CCL21 protein was 22.1% lower in comparison with DMEM medium as the control treatment (MTT assay was performed during 3 consecutive days with 3 different concentrations of recombinant CCL21). **B**: Cytotoxicity of commercial CCL21 protein at different concentrations (2.5, 5, 7.5, and 10 μg/ml) on MCF7 cancer cells was 28% lower in comparison with the control treatment, Also The IC50 of CCL21/IL1β plant recombinant protein against MCF7 cells was 3.86 μg/ml and IC50 of commercial CCL21 was 0.315 μg/ml. **C**: At a concentration of 7.5 μg/ml, survival rate of cancer cells incubated with non-recombinant protein was 74.3% lower compared to DMEM as control.

CCL21/IL1β recombinant protein and the non-recombinant protein. Therefore, we estimated that approximately 138 cells migrated within 24 hours (Fig 8B). For the other group, FBS 10% was placed (2.5 μg/mL) in the left well as the positive control, and DMEM medium as the negative control (2.5 μg/mL) in the right one, as depicted in Fig 8C. The result was the migration of 158 cells after 24 hours to the left well. The last group had two negative controls: one well was filled with non-recombinant protein, and the other with DMEM medium. Only a few cells (15 cells) were observed in each well, implying that neither chemoattractant factor had absorbed the cells (Fig 8D). An approximate quantification of the number of migrated cells was achieved using the Heit and Kubes method [48]. In this analysis, the number of migrated cells in the distance of wells containing cells and those containing chemoattractants was deduced from the number of cells that migrated between the wells containing cells and the wells containing non-absorbent material.Consequently, as shown in Fig 8, the number of monocytes moving toward the adsorbent substance was higher than that toward the non-adsorbent material, and the chemotaxis of the recombinant protein was estimated to be about 91% (Table 2). This modified

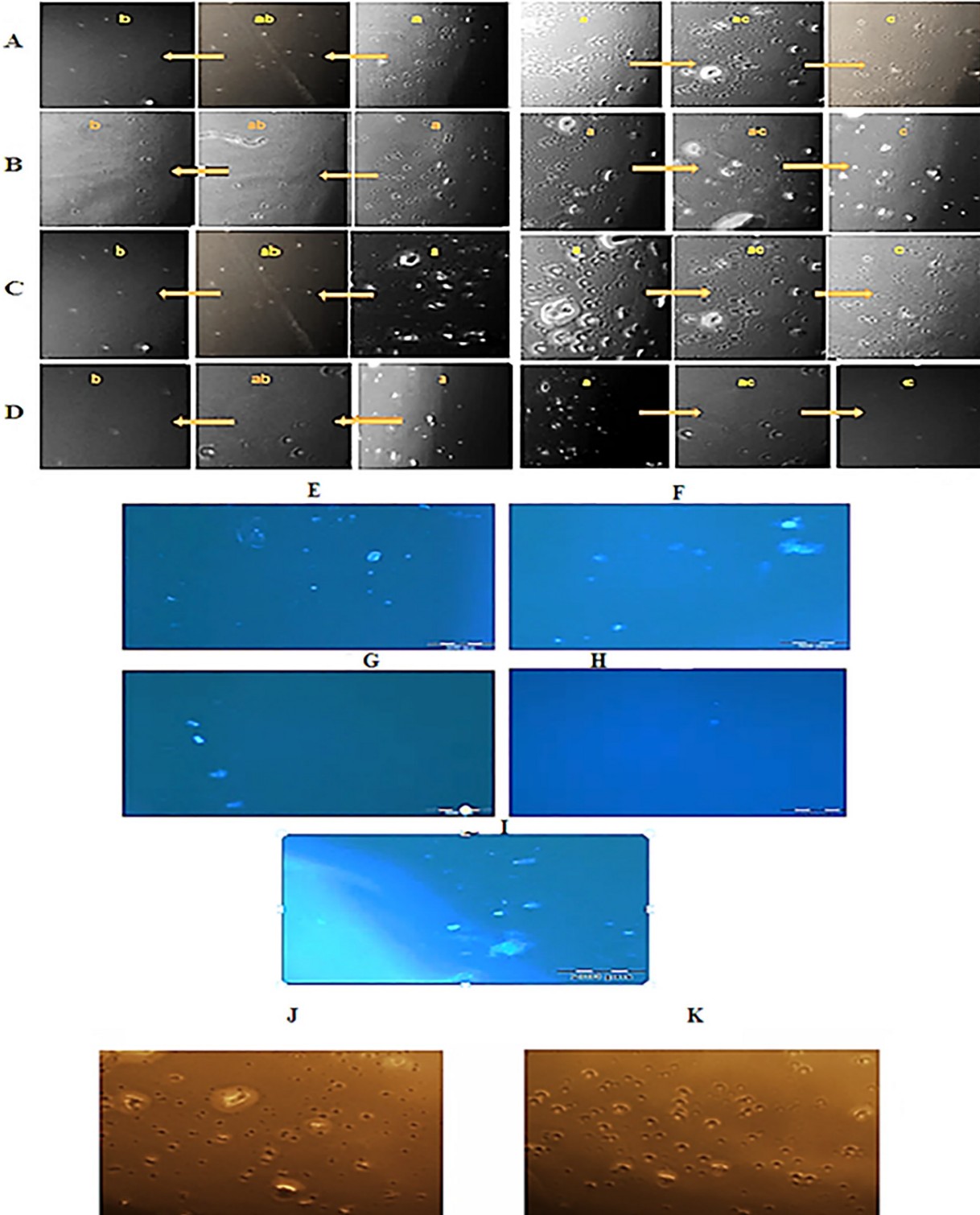

**Fig 8.** Cells (cells were seeded in the central well; ab: distance between the well containing cells and the well containing negative control (FBS-free DMEM); ac: distance between the well containing cells and the well containing chemoattractants such as CCL21 or FBS as positive control; c: well containing negative control (FBS-free DMEM); b: well containing chemoattractant (such as CCL21 or FBS) as positive control. **A)** Composite image of cells migrating from cell-containing well to the commercial CCL21 well (right) and the DMEM well (left). **B)** Composite image of cells migrating from cell-containing well to the purified CCL21 well (right) and the non-recombinant protein well (left). **C)** Composite image of cells

migrating from cell-containing well to 10% FBS well (right) and DMEM well (left). **D)** Composite image of cells migrating from cell-containing well to the non-recombinant protein well (right) and DMEM well (left). Chemotaxis of each substance was assessed by counting the number of cells migrated to each well. As can be seen, the number of monocyte cells moving toward the absorbent agent is higher than the non-absorbent substance, and chemotaxis of the recombinant protein is estimated to be about 91% (Bar = 200 μm). **E to I**: Nuclear labeling with DAPI staining. A black line represents the interface between agarose wall and the medium. Peripheral blood mononuclear cells (PBMC) migrated after 24 hours of stimulation toward **E)** commercial CCL21, **F)** purified CCL21, **G)** DMEM without 10% FBS, **H)** non-recombinant protein, and **I)** 10% FBS sources. **J&K**: Comparison of cell migration toward recombinant CCL21 protein as chemokinesis (CK) and toward growth factor as chemotaxis (CT). **J)** PBMC migration toward CCL21. **K)** PBMC migration toward 10% FBS. As shown, cells exhibited the same migration toward CCL21, the chemokine, and the growth factor (10% FBS), i.e. the adsorbent.

methodology allowed us to stain the nuclei of live cells by DAPI staining and use fluorescence microscopy to overcome the issue of low contrast between the agar background and the cells. However, in our experience, about 50% to 70% of the migrating cells might be lost due to cell staining (Fig 8E to 8I). Therefore, by examining the PBMC cells that had migrated toward the commercial CCL21, the recombinant plant CCL21, and 10% FBS (as the growth factor), we found that CCL21 is a chemokine with chemokinetic properties leading to the migration of immune cells. With this initial test, thus, it was possible to determine the role of CCL21/IL1β in migration and movement of immune cells towards tumor cells. Furthermore, comparison of chemotaxis (CT) and chemokinesis (CK) confirmed that there is no difference in cell migration toward CCL21 recombinant protein as chemokinesis and growth factor as chemotaxis was considered (Fig 8J and 8K).

## 4. Discussion

This research aimed at designing a codon-optimized *ccl21/IL1β* gene from human gene *ccl21* and the *interleukin-1 beta* adjuvant sequence VQGEESNDK, as well as examining the stability and binding affinity of this recombinant construct and CCR7 receptors through *in silico* analyses. The synthesized construct was introduced into tobacco plants, and expression, structure, and function of this recombinant protein were assessed. Since these two are human genes, and human is a very different organism from tobacco, the sequence was optimized for maximal protein expression in the selected host. Molecular dynamics simulation and various analyses were used to identify these two optimized genes [28]. Furthermore, ClusPro software was used to investigate the interaction between cytokine ligands and CCR7 receptors. The best cluster had a Z-Score of -2.3. The stability and flexibility of this recombinant protein were determined by MD simulation, and RMSD remained constant until the end of the simulation.

In SDS PAGE, dot blotting and western blotting assays, the purified protein sample from transgenic leaves generated a strong signal comparable to the positive control, whereas the protein from wild type plant was not detectable.

**Table 2. The approximate count of migrated cells and the percentage of chemotaxis according to Fig 7A to 7D [40].**

| Adsorbent substance | Number of cells in the well containing cells and absorbent material | Number of cells in the well containing cells and non-absorbent material | Chemotaxis rate/ Number of migrated cells | Chemotaxis percentage |
|---|---|---|---|---|
| **Commercial CCL21** (positive control as chemokine) | 170 | 17 | 153 | 90 |
| **Recombinant CCL21/IL1β** | 138 | 12 | 126 | **91** |
| **FBS10%** (positive control as chemotactic agent) | 158 | 17 | 141 | 89 |

Chemotaxis rate = (Number of cells in the well containing cells and absorbent material)–(Number of cells in the well containing cells and non-absorbent material)

Percentage of chomotoxicity = (Number of cells in the well containing cells and absorbent material/chemotaxis rate) × 100 [43]

In previous studies on expression of other recombinant proteins in tobacco plants, TSP yields of 0.019–0.31% were reported [49, 50]. In this work, however, CCL21/IL1β was expressed in the transgenic tobacco lines at levels up to 1.81–2.14% of TSP, and also ELISA results demonstrated that CCL21/IL1β has been correctly expressed and folded in tobacco plants and was fully functional.

FTIR was used to investigate the structure of the recombinant protein and to ensure the insertion and expression of the foreign gene into the tobacco plants. FTIR assessment confirmed the presence of the recombinant protein (and thus the synthetic construct) in transgenic leaves of tobacco. Commercial CCL21 had a similar peak profile in most of the absorption areas of the spectra. Put together, these two sets of data indicate that both proteins had a similar molecular structure even though there was a high dissimilarity in peak positions between transgenic and non-transgenic tobacco. FTIR measurement and MALDI-TOF/TOF mass spectrum assessments confirmed that *ccl21/IL1β* was correctly expressed in the tobacco plant.

We also investigated the potential of the purified recombinant protein for stimulating the proliferation and migration of cancer cells by the wound healing method [45, 51, 52]. Cancer cell lines migrated into the wound site 24 hours after seeding cells in the MCF7 growth medium (15% FBS). However, after replacing the growth media with a medium with different CCL21/IL1β protein treatments, the purified protein extract from transgenic leaves and the commercial protein remarkably prevented the migration and growth of cancer cell lines in monolayer cell cultures compared to control groups (basal medium and non-recombinant protein). Furthermore, the results demonstrated a decrease in the survival rate and metastasization of cancer cells in the presence of CCL21 recombinant protein.

The MTT assay was utilized to evaluate the cytotoxicity of CCL21/IL1β on cancer cells. In this method, the yellow tetrazolium MTT (3-(4, 5-dimethyl thiazolyl-2)-2, 5-diphenyltetrazolium bromide) is reduced by metabolically active cells. The MTT assay measures the cell proliferation rate, and cytotoxicity leads to reduced cell viability and apoptosis [53]. MTT results showed that the $IC_{50}$ of CCL21/IL1β on MCF7 cells was less than that of the non-recombinant protein. In addition, toxicity of commercial CCL21 on MCF7 cells was similar to CCL21/IL1β recombinant protein, which is the anticancer antigen used for lung cancer treatment [54]. Monitoring CCL21/IL1β activity for 72 hours showed that the two components act in a dose and time-dependent manner. Chemotactic abilities could be measured using different criteria, such as the number of attracted cells, induced speed, and morphological features [55]. The chemokinetic activity of this recombinant protein in stimulating cell motility and chemotactic movement of PBMC cell populations was investigated by agarose assay. Agarose assay has been used for studying cellular chemotactic reactions and the signaling processes underlying them [55, 56]. Generally speaking, chemokinesis is a chemically induced motile response of unicellular organisms to chemical materials that proved the cells to alter their migratory behaviors. However, chemokinesis has a a more random nature compared to chemotaxis [48]. Hence, the results of our study demonstrate a differentiation between chemokinetic (CCL21) and chemotactic (FBS) movement. It is expected that this recombinant multi-epitope protein can be considered for cancer treatment. It can also treat viral diseases, such as AIDS and coronaviruses (e.g. Covid-19) since it triggers the immune system to produce potent antibodies in patients with these diseases.

## 5. Conclusion

Chemotherapy is a method mainly used for treating cancer, which unfortunately causes adverse side effects such as hair loss, nausea, and weakness. complications occur because

cancer treatment aims to kill cancer cells, and some of the body's healthy cells are also damaged in the process. For example, one of the drugs prescribed to most cancer patients is Herceptin which, in addition to its high cost, weakens the immune system and causes many side effects. Recombinant proteins, however, can be produced at a minimal cost and have no particular side effects, strengthening the immune system and immunizing the patient before chemotherapy. In addition, many of the drugs that have been suggested for treatment of viral diseases, such as AIDS and coronaviral conditions, produce potent antibodies in the body, which can stimulate the immune system of the patient. So far, many commercial drugs have been designed and produced for stimulating and strengthening the patient's immune system. Commercial drugs based on CCL21 are also available in the market. The International Cancer Institute now recognizes CCL21 as an effective drug in the treatment of AIDS and cancer. However, these drugs have limitations in immunotherapy. For example, CCL21 only affects T-lymphocytes and CTLs and reduces regulatory T-cells (Tregs) expression and has no role in B-lymphocytes, macrophages, and neutrophils. The recombinant drug produced in this study does not have any special chemical side effects and will certainly not cost much if it is commercialized. If successful in the clinical phase. To the best of our knowledge, this study is the first to report expression of *ccl21/IL1β* construct in tobacco via agrobacterium gene transformation method and high-level expression of CCL21/IL1β. The cost of producing this recombinant drug is very low compared to the commercial antigen. The experiments confirmed that the recombinant protein has epitope sequences covering more than 90% of the HLA-DRB4 allele. Therefore, this recombinant protein is presented by APCs after binding and endocytosed through the CCR7 receptor. This antigen-receptor complex induces efficient CD4+ T-cell responses by the MHC class I and II antigen processing pathways. Furthermore, it is a potential option in the immunotherapy of cancer.

## Supporting information

**S1 Raw image.**
(PDF)

## Acknowledgments

We would like to thank Stem Cells and Regenerative Medicine Research Group, the Academic Center for Education, Culture, and Research (ACECR), Razavi Khorasan Branch, Mashhad, Iran.

## Author Contributions

**Conceptualization:** Maria Beihaghi.

**Data curation:** Maria Beihaghi, Samad Khaksar.

**Formal analysis:** Maria Beihaghi, Masoud Chaboksavar, Samad Khaksar.

**Funding acquisition:** Maria Beihaghi.

**Investigation:** Maria Beihaghi.

**Methodology:** Maria Beihaghi, Samad Khaksar.

**Project administration:** Hasan Marashi, Maria Beihaghi, Homan Tehrani, Ardavan Abiri.

**Resources:** Maria Beihaghi.

**Software:** Maria Beihaghi, Masoud Chaboksavar, Ardavan Abiri.

**Supervision:** Hasan Marashi.

**Validation:** Maria Beihaghi.

**Visualization:** Maria Beihaghi.

**Writing – original draft:** Maria Beihaghi, Samad Khaksar, Homan Tehrani.

**Writing – review & editing:** Maria Beihaghi, Samad Khaksar, Homan Tehrani.

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
