## [Decision Letter · Decision Letter 0]

17 Jan 2022

PONE-D-21-35241In silico analysis and in planta production of recombinant ccl21/IL1β protein and characterization of its in vitro anti-tumor and immunogenic activityPLOS ONE

Dear Dr. Beihaghi,

Thank you for submitting your manuscript to PLOS ONE. After careful consideration, we feel that it has merit but does not fully meet PLOS ONE’s publication criteria as it currently stands. Therefore, we invite you to submit a revised version of the manuscript that addresses the points raised during the review process. Please submit your revised manuscript by Mar 03 2022 11:59PM. If you will need more time than this to complete your revisions, please reply to this message or contact the journal office at plosone@plos.org. Please include the following items when submitting your revised manuscript:A rebuttal letter that responds to each point raised by the academic editor and reviewer(s). You should upload this letter as a separate file labeled 'Response to Reviewers'.A marked-up copy of your manuscript that highlights changes made to the original version. You should upload this as a separate file labeled 'Revised Manuscript with Track Changes'.An unmarked version of your revised paper without tracked changes. You should upload this as a separate file labeled 'Manuscript'.

We look forward to receiving your revised manuscript.

Kind regards,

Irina V. Balalaeva, PhD

Academic Editor

PLOS ONE

Journal Requirements:

2. Thank you for stating the following in the Acknowledgments/Funding Section of your manuscript:

Funding:

“The research was financed by the Office of Vice President for Research and Technology of Ferdowsi University of Mashhad, Iran, Stem Cells and Regenerative Medicine Research Group, the Academic Center for Education, Culture, and Research.”

Acknowledgement:

“This study was financially supported by the Biotechnology Development Council of the Islamic Republic of Iran.”

We note that you have provided additional information within the Acknowledgements/Funding Section that is not currently declared in your Funding Statement. Please note that funding information should not appear in the Acknowledgments section or other areas of your manuscript. We will only publish funding information present in the Funding Statement section of the online submission form.

“The author(s) received no specific funding for this wor”

3. PLOS requires an ORCID iD for the corresponding author in Editorial Manager on papers submitted after December 6th, 2016. Please ensure that you have an ORCID iD and that it is validated in Editorial Manager. To do this, go to ‘Update my Information’ (in the upper left-hand corner of the main menu), and click on the Fetch/Validate link next to the ORCID field. This will take you to the ORCID site and allow you to create a new iD or authenticate a pre-existing iD in Editorial Manager. Please see the following video for instructions on linking an ORCID iD to your Editorial Manager account: https://www.youtube.com/watch?v=_xcclfuvtxQ.

Reviewers' comments:

Reviewer's Responses to Questions

**Comments to the Author**

1. Is the manuscript technically sound, and do the data support the conclusions?

Reviewer #1: Partly

Reviewer #2: No

2. Has the statistical analysis been performed appropriately and rigorously? 

Reviewer #1: No

Reviewer #2: I Don't Know

3. Have the authors made all data underlying the findings in their manuscript fully available?

Reviewer #1: No

Reviewer #2: No

4. Is the manuscript presented in an intelligible fashion and written in standard English?

Reviewer #1: Yes

Reviewer #2: No

5. Review Comments to the Author

Reviewer #1: In the current manuscript entitled “In silico analysis and in planta production of recombinant ccl21/IL1β protein and characterization of its in vitro anti-tumor and immunogenic activity” (Manuscript ID: PONE-D-21-35241) by Beihaghi et al. investigated the production of a codon-optimized fusion protein in plants and conducted in silico analysis and functional testing. They used an MTT assay to assess the cell growth inhibiting properties of the protein and compared it to a commercial and native counterpart.

They have submitted their manuscript for publication in PLOS ONE.

The topic is of interest to researchers in the biopharmaceutical and plant biotechnology area. The manuscript is well structured and mostly well written. The experiments appear well designed and executed. However, there are some limitations due to which I think that the text is not yet ready for publication in PLOS ONE:

1. Several references are missing

2. Details in the methods section are missing

3. Statistical analyses and result quantification is missing

4. The degree of novelty is rather low, especially compared to the authors own previous publications, e.g. https://www.ncbi.nlm.nih.gov/pmc/articles/PMC5730375/

I have prepared a detailed list of comments in an annotated pdf.

Due to the interesting results the authors present, I encourage them to revise the manuscript accordingly and submit an updated version.

Reviewer #2: The manuscript represent an interesting piece of work which in principle is publishable. However, it requires significant modification in order to make it suitable for publication. Therefore, my recommendation is MAJOR REVISION. After the authors have completed these revisions, the new manuscript should undergo further peer review.

1- I guess my main concern about the manuscript is that in no place has the sequence of the chimeric CCL21/IL1beta protein spelt out. CCL21 interacts with its receptor through both the the N-terminus, and the main sequence whilst it’s C-terminus is mostly responsible for interaction with glucosaminoglycans on the surface of endothelial cells. Depending on where the IL1beta sequence is inserted, these interactions can significantly affect the molecular interactions of the peptide with its receptor, CCR7 (which is the point of the paper). I presume the purpose of the molecular modelling was to demonstrate that these interactions still work, although that is by no means clear. I am also unclear about the size of the protein. On line 452, the authors state that they expected a MW of 55kDa (although they don’t explain why) which is quite large for a cytokine (CCL21 and IL1b have MW of 12.5 and 17.5 kDa respectively). However, figure 1B doesn’t look that different from the CCL21 WT. How is that possible? (see also comment 10)

2- That omission, is only one example of the wider issue with the manuscript. It is highly unfocused and contained on occasion superfluous information, and on other occasion not enough information. For example, section 2.1.2 doesn’t really add anything to the manuscript. On the other hand in line 379-380 the authors state one of the 10 models generated was used without mentioning what the selection criteria was.

3- There are quite a large number of errors and omission in the manuscript. These are far too many for me to list but as an example:

4- In line 459, the authors mention the sequence VQEESNDK from IL1b and mass is given as 947.1 Da, which is correct for that sequence. This is used as proof of inclusion of IL1b. However, according to the authors themselves in lines 95 and 363 (and also according to Uniprot) the correct sequence is V(162)QGEESNDK(171) with a mass of 1004 Da. Can the authors explain this discrepancy?

5- Line 363 “DNA sequences encoding human CCL21 used in this structure have a DCCL motif…”

6- Line 365 “Since their IC50 is less than 50” (no unit given) etc etc etc

7- I recommend the authors do not use “CCR7+” when refer to CCR7 expressing immortalised cell line, MCF-7. The terminology is widely used for dendritic cells and T cells where gain of expression is associated with gain of functional capabilities of the cells. For immortalised cell lines, “+” is often used in the context of knock in, so it’s use can be confusing.

8- Related to the above, although there are no strict guidelines for this journal, I strongly recommend to the editor that the authors should provide an STR based certificate of authentication for the MCF-7 cell line used in this study. (Even though there is no doubt MCF-7 do express CCR7, see https://pubmed.ncbi.nlm.nih.gov/21151474/).

9- Generally speaking the quality of the figures is very poor. The link for reviewers did not contain the original figures.

10- In the manuscript, the in silico data for binding of CCL21-IL1b chimera with CCR7 is purely visual. Obviously this is not sufficient information to support the key hypothesis of the manuscript, so authors need to provide more information on the interactions between the chimera and receptor at a molecular level.

6. PLOS authors have the option to publish the peer review history of their article (what does this mean?). If published, this will include your full peer review and any attached files.

Reviewer #1: No

Reviewer #2: No

---

## [Author Response · Author response to Decision Letter 0]

1 Apr 2022

Dear Reviewers, 

Thank a lot you for your consideration and all of your comments, I have noticed them and highlight them in manuscript also we have increase the resolution of figures and some figures was added and highlight manuscript have been uploaded also I have been attached some original figures and upload it please find attached files

Best Regard

Maria Beihaghi

---

## [Decision Letter · Decision Letter 1]

18 Apr 2022

PONE-D-21-35241R1In silico analysis and in planta production of recombinant ccl21/IL1β protein and characterization of its in vitro anti-tumor and immunogenic activityPLOS ONE

Dear Dr. Beihaghi,

Thank you for submitting your manuscript to PLOS ONE. After careful consideration, we feel that it has merit but does not fully meet PLOS ONE’s publication criteria as it currently stands. Therefore, we invite you to submit a revised version of the manuscript that addresses the points raised during the review process.

We look forward to receiving your revised manuscript.

Kind regards,

Irina V. Balalaeva, PhD

Academic Editor

PLOS ONE

Reviewers' comments:

Reviewer's Responses to Questions

**Comments to the Author**

1. If the authors have adequately addressed your comments raised in a previous round of review and you feel that this manuscript is now acceptable for publication, you may indicate that here to bypass the “Comments to the Author” section, enter your conflict of interest statement in the “Confidential to Editor” section, and submit your "Accept" recommendation.

Reviewer #1: (No Response)

2. Is the manuscript technically sound, and do the data support the conclusions?

Reviewer #1: Partly

3. Has the statistical analysis been performed appropriately and rigorously? 

Reviewer #1: No

4. Have the authors made all data underlying the findings in their manuscript fully available?

Reviewer #1: No

5. Is the manuscript presented in an intelligible fashion and written in standard English?

Reviewer #1: Yes

6. Review Comments to the Author

Reviewer #1: I thank the authors for submitting a revised version of the manuscript. However, when checking the comments I had shared with them in an annotated pdf in the first round of revision, I found that most of them have not been addressed. The comments that were addressed mostly dealt with the addition of links. I will not list in detail which comments have not been addressed, but ask that the authors double check the annotated pdf and do so on their own. I would just like to give some examples here:

1. three points were raised in the abstract, none was addressed

2. I had asked for details about greenhouse conditions, but not additional information is provided

3. I had asked to consistently use mass instead of weight, but the authors introduce units such as "w/v", where "w" is obliviously "weight"

4. In section 2.11 I had asked for some details on the protocol, but the authors provide text book knowledge about chemotaxis

5. In section 3.1.3 the authors had mentioned different models they tested and had asked how they did that etc. Instead of adding information, that section was removed without a comment

etc.

Therefore, I still think that the manuscript is not yet ready for publication and the authors should really address the comments one-by-one as stated in their response.

7. PLOS authors have the option to publish the peer review history of their article (what does this mean?). If published, this will include your full peer review and any attached files.

Reviewer #1: No

---

## [Author Response · Author response to Decision Letter 1]

6 May 2022

Dear Reviewer Thanks a lot for all of your comments, But most of your comments was changed in manuscript track changes file that have been sent in revise submission file before. So I have noticed point by point and add some paragraph and highlight them in manuscript track changes file, Also the other comment was mentioned and noticed in PONE-D-21-35241_reviewer_comments-3.pdf file.

Best Regard

Maria Beihaghi

---

## [Decision Letter · Decision Letter 2]

15 Jun 2022

PONE-D-21-35241R2In silico analysis and in planta production of recombinant ccl21/IL1β protein and characterization of its in vitro anti-tumor and immunogenic activityPLOS ONE

Dear Dr. Beihaghi,

Thank you for submitting your manuscript to PLOS ONE. After careful consideration, we feel that it has merit but does not fully meet PLOS ONE’s publication criteria as it currently stands. Therefore, we invite you to submit a revised version of the manuscript that addresses the points raised during the review process.

We look forward to receiving your revised manuscript.

Kind regards,

Irina V. Balalaeva, PhD

Academic Editor

PLOS ONE

Journal Requirements:

Reviewers' comments:

Reviewer's Responses to Questions

**Comments to the Author**

1. If the authors have adequately addressed your comments raised in a previous round of review and you feel that this manuscript is now acceptable for publication, you may indicate that here to bypass the “Comments to the Author” section, enter your conflict of interest statement in the “Confidential to Editor” section, and submit your "Accept" recommendation.

Reviewer #1: (No Response)

2. Is the manuscript technically sound, and do the data support the conclusions?

Reviewer #1: Partly

3. Has the statistical analysis been performed appropriately and rigorously? 

Reviewer #1: No

4. Have the authors made all data underlying the findings in their manuscript fully available?

Reviewer #1: Yes

5. Is the manuscript presented in an intelligible fashion and written in standard English?

Reviewer #1: Yes

6. Review Comments to the Author

Reviewer #1: I thank the authors for submitting a second revised version of the manuscript and for responding to my comments in the annotated pdf. However, I feel that the responses are inadequate in several instances. For example:

1. the abstract has to be self-explanatory. Therefore, responding that, for example, the wound healing method, is explained in the main text appears to be insufficient.

2. the description of the statistical analysis is insufficient, e.g. which tests were used to establish normality of the data before applying Duncan’s test?

3. why was the defensine linker used? The authors claim that this is explained in lines 31-57 (which is rather vague), but this linker is not mentioned at all in this section.

etc.

Therefore, I still think that the manuscript is not yet ready for publication and the authors should double check the comments with their previous response and edits in the manuscript.

Also, I kindly ask that the authors revise the language of the manuscript with the assistance of a native speaker.

7. PLOS authors have the option to publish the peer review history of their article (what does this mean?). If published, this will include your full peer review and any attached files.

Reviewer #1: No

---

## [Author Response · Author response to Decision Letter 2]

12 Jul 2022

Dear Reviewer Thanks a lot for all of your comments, I have noticed point by point and highlight them in manuscript.

---

## [Editor Report · Decision Letter 3]

21 Jul 2022

In silico analysis and in planta production of recombinant ccl21/IL1β protein and characterization of its in vitro anti-tumor and immunogenic activity

PONE-D-21-35241R3

Dear Dr. Beihaghi,

We’re pleased to inform you that your manuscript has been judged scientifically suitable for publication and will be formally accepted for publication once it meets all outstanding technical requirements.

Kind regards,

Irina V. Balalaeva, PhD

Academic Editor

PLOS ONE
---

## [Editor Report · Acceptance letter]

17 Aug 2022

PONE-D-21-35241R3 

*In silico* analysis and *in planta* production of recombinant *ccl21/IL1β* protein and characterization of its *in vitro* anti-tumor and immunogenic activity 

Dear Dr. Beihaghi:

I'm pleased to inform you that your manuscript has been deemed suitable for publication in PLOS ONE. Congratulations! Your manuscript is now with our production department. 

Kind regards, 

on behalf of

Dr. Irina V. Balalaeva 

Academic Editor

PLOS ONE